# Efficient Online Estimation of Causal Effects by Deciding What to Observe

**Shantanu Gupta, Zachary C. Lipton, David Childers**

Carnegie Mellon University
{shantang,zlipton,dchilders}@cmu.edu

## Abstract

Researchers often face data fusion problems, where multiple data sources are available, each capturing a distinct subset of variables. While problem formulations typically take the data as given, in practice, data acquisition can be an ongoing process. In this paper, we aim to estimate any functional of a probabilistic model (e.g., a causal effect) as efficiently as possible, by deciding, at each time, which data source to query. We propose *online moment selection* (OMS), a framework in which structural assumptions are encoded as moment conditions. The optimal action at each step depends, in part, on the very moments that identify the functional of interest. Our algorithms balance exploration with choosing the best action as suggested by current estimates of the moments. We propose two selection strategies: (1) explore-then-commit (OMS-ETC) and (2) explore-then-greedy (OMS-ETG), proving that both achieve zero asymptotic regret as assessed by MSE. We instantiate our setup for average treatment effect estimation, where structural assumptions are given by a causal graph and data sources may include subsets of mediators, confounders, and instrumental variables.

## 1 Introduction

Statistical and causal modeling typically proceed from the assumption that we already know which variables are (and are not) observed. However, this perspective fails to address the difficult data collection decisions that precede such modeling efforts. Doctors must select a set of tests to run. Survey designers must select a slate of questions to ask. Companies must select which datasets to purchase. Whether or not we model these decisions, they pervade the practice of data science, influencing both what questions we can ask and how accurately we can answer them.

One might ask, *why not collect everything?* The answers are two-fold: First, data acquisition can be expensive. In a medical setting, blood tests can cost anywhere from tens to thousands of dollars. Running every test for every patient is infeasible. Likewise, space on surveys is limited, and asking every conceivable question of every respondent is infeasible. Second, in many settings, we lack complete control over the set of variables observed. Instead, we might have access to multiple data sources, each capturing a different subset of variables. Such *data fusion problems* pervade economic modeling and public health, and present interesting challenges: (i) efficiently estimating (or even identifying) a population parameter of interest often requires intelligently combining data from multiple sources; (ii) data collection is often iterative, with tentative conclusions at each stage informing choices about what data to collect next.

In this paper, we formalize the sequential problem of deciding, at each time, which data source to query (i.e., *what to observe*) in order to efficiently estimate a target parameter. We propose online moment selection (OMS), a framework that applies the generalized method of moments (GMM) [17] both to estimate the parameter and to decide which data sources to query. This framework can be

35th Conference on Neural Information Processing Systems (NeurIPS 2021).

applied to estimate any statistical parameter that can be identified by a set of moment conditions. For example, OMS can address (i) any (regular) maximum likelihood estimation problem [13, Page 109]; and (ii) estimating average treatment effects (ATEs) using instrumental variables (IVs), backdoor adjustment sets, mediators, and/or other identification strategies.

Our strategy requires only that the agent has sufficient structural knowledge to formulate the set of moment conditions and that each moment can be estimated using the variables returned by at least one of the data sources. Interestingly, the optimal decisions which lead to estimates with the lowest mean squared error (MSE) depend on the (unknown) model parameters. This motivates our adaptive strategy: as we collect more data, we better estimate the underlying parameters, improving our strategy for allocating our remaining budget among the available data sources.

We first address the setting where the cost per instance is equal across data sources (Section 4). First, we show that any fixed policy that differs from the oracle suffers constant asymptotic regret, as assessed by MSE. We then overcome this limitation by proposing two adaptive strategies—explore-then-commit (OMS-ETC) and explore-then-greedy (OMS-ETG)—both of which choose data sources based on the *estimated* asymptotic variance of the target parameter.

Under OMS-ETC, we use some fraction of the sample budget to explore randomly, using the collected data to estimate the model parameters. We then exploit the current estimated model, collecting the remaining samples according to the fraction expected to minimize our estimator's asymptotic variance. In OMS-ETG, we continue to update our parameter estimates after every step as we collect new data. We prove that both policies achieve zero asymptotic regret. To overcome the non-i.i.d. nature of the sample moments, we draw upon martingale theory. To derive zero asymptotic regret, we show uniform concentration of sample moments and a finite-sample concentration inequality for the GMM estimator with dependent data. Next, we adapt OMS-ETC and OMS-ETG to handle heterogeneous costs over the data sources (Section 5) and prove that they still have zero asymptotic regret.

Finally, we validate our findings experimentally [1] (Section 6). Motivated by ATE estimation in causal models encoded as directed acyclic graphs, we generate synthetic data from a variety of causal graphs and show that the regret of our proposed methods converges to zero. Furthermore, we see that despite being asymptotically equivalent, OMS-ETG outperforms OMS-ETC in finite samples. Finally, we demonstrate the effectiveness of our methods on two semi-synthetic datasets: the Infant Health Development Program (IHDP) dataset [19] and a Vietnam era draft lottery dataset [2].

## 2  Related Work

Many works attempt to identify and estimate causal effects from multiple datasets. [4, 21] study the problem of combining multiple heterogeneous datasets and propose methods for dealing with various biases. Other works study causal identification when observational and interventional distributions involving different sets of variables are available [28, 40]. [12] introduce estimators of the ATE that efficiently combine two datasets, one where confounders are observed (enabling the backdoor adjustment) and another where mediators are observed (enabling the frontdoor adjustment). [29] derive an estimator for the ATE in linear causal models with multiple confounders, where the confounders are observed in different datasets.

Another related line of work addresses finding optimal adjustment sets for covariate adjustment [18, 33, 45, 36]. While these works take for granted the collection of available datasets, we focus on the problem of deciding which data to collect. Our work shares motivation with active learning, where the learner strategically chooses which (unlabeled) samples to label in order to learn most efficiently [35, 26]. [8] design algorithms for actively collecting samples in a manner that minimizes the learner's variance. In settings where there is a cost associated with collecting each feature, active feature acquisition methods incrementally query feature values to improve a predictive model [22, 34, 20]. [47] propose an active learning criterion to find the most informative questions to ask each respondent in a survey. In the context of causal inference, [37] study active structure learning of causal DAGs by finding cost-optimal interventions, [11] demonstrate that for ATE estimation, actively deconfounding data can improve sample complexity. [44] propose strategies for acquiring missing confounders to efficiently estimate the ATE.

---

[1]The code and data are available at `https://www.github.com/acmi-lab/online-moment-selection`.

Others have studied moment selection and IV selection from batch data. [1] introduce consistent moment selection procedures for the GMM setting with some incorrect moments. [6] propose an information-based lasso method for excluding invalid or redundant moment conditions. [41] propose a variable selection framework that uses lasso regression to decide which covariates to include. [14] propose four statistical criteria—including estimation efficiency and non-redundancy—for selecting among a set of candidate IVs. [9] develop an IV selection criteria based on asymptotic MSE and develop it for the GMM and generalized empirical likelihood estimators. [7] develop conservative confidence intervals for structural parameters when the off-diagonal entries of the covariance matrix of the empirical moments are unknown (e.g., when the moments are obtained from different datasets). By contrast, we are interested in selecting the data sources for the moments in an online setting.

Previous works address learning from adaptively collected data. [25] propose methods for adaptive experimental design, where at each step, the experimenter must decide the treatment probability using past data in order to efficiently estimate the ATE. [23, 24] propose a doubly-robust estimator for off-policy evaluation with dependent samples. [46] provide regret bounds for learning an optimal policy using adaptively collected data, where the probability of selecting an action is a function of past data. [48, 49] study statistical inference for OLS and M-estimation with non-i.i.d. bandit data. While these settings are different from ours, some of the theoretical tools (e.g., martingale asymptotics and uniform martingale concentration bounds) are similar.

## 3   Preliminaries

In the GMM framework [17, 30], we estimate model parameters by leveraging moment conditions that are satisfied at the true parameters $\theta^*$. A moment condition is a vector $g(X_t, \theta)$ such that $\mathbf{E}[g(X_t, \theta^*)] = 0$. We estimate $\theta^*$ by minimizing the objective $\widehat{Q}_T$:

$$\widehat{\theta}_T = \arg\min_{\theta \in \Theta} \widehat{Q}_T(\theta), \quad \text{where} \quad \widehat{Q}_T(\theta) = \left[\frac{1}{T}\sum_{t=1}^{T} g_t(\theta)\right]^{\top} \widehat{W} \left[\frac{1}{T}\sum_{t=1}^{T} g_t(\theta)\right],$$

$\Theta$ is the parameter space, $g_t(\theta) := g(X_t, \theta)$, and $\widehat{W}$ is some (possibly data dependent) positive definite matrix. In this work, we use the *two-step GMM estimator*, where the *one-step estimator* $\widehat{\theta}_T^{(\text{os})}$ is computed with $\widehat{W} := I$ (identity) and the two-step estimator with $\widehat{W} := \left[\widehat{\Omega}_T(\widehat{\theta}_T^{(\text{os})})\right]^{-1}$, where $\widehat{\Omega}_T(\theta) = \left[\frac{1}{T}\sum_{t=1}^{T} g_t(\theta) g_t(\theta)^{\top}\right]$.

Let $\mathbf{V}$ be the set of variables of interest and $\psi$ a collection of subsets of $\mathbf{V}$, each corresponding to the specific variables observable via one of the available data sources. Our methods are applicable whenever the target parameter can be identified by a set of moment conditions such that each moment relies on variables simultaneously observable in at least one data source. The *selection vector*, denoted by $s_t \in \{0, 1\}^{|\psi|}$, is the binary vector indicating the data source selected at time $t$.

**Assumption 1.** *The agent queries one data source at each step:* $\sum_{i=1}^{|\psi|} s_{t,i} = 1$, *i.e., $s_t$ is one-hot.*

We can handle the querying of multiple sources by adding the union of their variables to $\psi$. In our setup, the moment conditions can be written as $g_t(\theta) = m(s_t) \otimes \tilde{g}_t(\theta) \in \mathbf{R}^M$, where $\otimes$ is the Hadamard product, $m : \{0, 1\}^{|\psi|} \to \{0, 1\}^M$ is a fixed known function such that $m(s_t)$ determines which moments get selected, and $\tilde{g}_t(\theta)$ are i.i.d. across $t$. For concreteness, we instantiate our setup with a simple example:

**Example 1** (Instrumental Variable (IV) graph). *Consider a linear IV causal model (Figure 2a) with instrument $Z$, treatment $X$, outcome $Y$, and the following data-generating process:*

$$X := \alpha Z + \eta, \quad Y := \beta X + \epsilon, \quad \epsilon \not\!\perp\!\!\!\perp \eta, \quad \epsilon \perp\!\!\!\perp Z, \quad \eta \perp\!\!\!\perp Z.$$

*The target parameter is the ATE $\beta$. For $\psi = \{\{Z, X\}, \{Z, Y\}\}$, the moment conditions are*

$$g_t(\theta) = \underbrace{\begin{bmatrix} s_{t,1} \\ s_{t,2} \end{bmatrix}}_{=m(s_t)} \otimes \underbrace{\begin{bmatrix} Z_t(X_t - \alpha Z_t) \\ Z_t(Y_t - \alpha\beta Z_t) \end{bmatrix}}_{=\tilde{g}_t(\theta)} = \begin{bmatrix} s_{t,1} Z_t(X_t - \alpha Z_t) \\ (1 - s_{t,1}) Z_t(Y_t - \alpha\beta Z_t) \end{bmatrix},$$

*where $\theta = [\beta, \alpha]^{\top}$ and $\{Z_t, X_t, Y_t\}$ are i.i.d.*

For some known function $f_{\text{tar}} : \Theta \to \mathbf{R}$, let $\beta^* := f_{\text{tar}}(\theta^*)$ be the target parameter (e.g., the ATE). In practice, we estimate the target parameter by plugging-in the GMM estimate: $\widehat{\beta} = f_{\text{tar}}(\widehat{\theta})$. Let $H_t$ represent the *history* or the data collected until time $t$ with $H_0 = \{\}$ and space $\mathcal{H}_t$. A data collection policy $\pi$ consists of a sequence of functions $\pi_t : \mathcal{H}_{t-1} \to \{0, 1\}^{|\psi|}$ with $s_t = \pi_t(H_{t-1})$. Thus $s_t$ can be dependent on data collected until time $(t-1)$ and so the sample moments $g_t(\theta)$ are *not* i.i.d.

**Definition 1** (Selection ratio). *The selection ratio, denoted by $\kappa_t^{(\pi)}$, encodes the fraction of samples collected from each data source until time $t$: $\kappa_t^{(\pi)} = \frac{1}{t} \sum_{i=1}^{t} s_t \in \Delta^{|\psi|-1}$ (standard simplex).*

We use $\widehat{\theta}_t^{(\pi)}$ and $\widehat{\theta}_t^{(\text{os})}$ to denote the two-step and one-step GMM estimators, respectively, that use the data $H_t$. To reduce clutter, we use $\Delta_\psi := \Delta^{|\psi|-1}$, $\text{ctr}(\Delta_\psi) = \left[ \frac{1}{|\psi|}, \frac{1}{|\psi|}, \ldots, \frac{1}{|\psi|} \right]$ (center of the simplex), and might drop the superscript $\pi$ from $\kappa_t$, $\widehat{\theta}_t$, and $\widehat{\beta}_t$. $\|.\|$ denotes the spectral and $l_2$ norms for matrices and vectors, respectively, and $N_\epsilon(\theta) := \{\theta' : \|\theta' - \theta\| \le \epsilon\}$ ($\epsilon$-ball around $\theta$).

## 4 Adaptive Data Collection

The central challenge in this work is to make strategic decisions about which data to observe so that we most efficiently estimate the target functional. In this section, we present three policies: (i) *fixed*: query the data sources according to a pre-specified ratio; (ii) OMS-ETC: query uniformly for a specified exploration period, estimate the optimal ratio based on the inferred parameters and thereafter aim for that ratio; and (iii) OMS-ETG: same as OMS-ETC but continue to update parameter (and thus oracle ratio) estimates after the exploration period. For now, we analyze these policies for the case when the cost to query is identical across data sources. By $T \in \mathbb{N}$, we denote the (known) *horizon*, which can be thought of as the agent's data acquisition budget. We defer all proofs to Appendix A.

We now present sufficient conditions for consistency and asymptotic normality of the GMM estimator under adaptively collected data. We later use these results to derive the regret of our policies.

**Assumption 2.** *(a) (Identification) $\forall \theta \ne \theta^*, \mathbf{P}\left(\liminf_{T \to \infty} \bar{Q}(\theta) > 0\right) = 1$, where $\bar{Q}(\theta) = g_T^*(\theta)^\top \widehat{W} g_T^*(\theta)$ and $g_T^*(\theta) = \left[ \frac{1}{T} \sum_{t=1}^{T} m(s_t) \otimes \mathbf{E}\left[ \tilde{g}_t(\theta) \right] \right]$; (b) $\Theta \subset \mathbf{R}^D$ is compact; (c) $\forall \theta, \tilde{g}_i(\theta)$ is twice continuously differentiable (c.d.); (d) $\forall \theta, \mathbf{E}[\tilde{g}_i(\theta)]$ is continuous; and (e) $f_{tar}$ is c.d. at $\theta^*$.*

By Assumption 2(a), the GMM objective is uniquely minimized at $\theta^*$. Informally, this means that each moment is (asymptotically) collected enough times to allow identification. If $M = D$ (just-identified case), this holds when (i) an asymptotically non-negligible fraction of every moment is collected: $\forall j \in [M]$, $\mathbf{P}\left(\liminf_{T \to \infty} \frac{1}{T} \sum_{t=1}^{T} m_j(s_t) \ne 0\right) = 1$; and (ii) $\forall \theta \ne \theta^*$, $\mathbf{E}\left[\tilde{g}_t(\theta)\right] \ne 0$.

**Property 1** (ULLN). *Let $a_i(\theta) := a(X_i; \theta) \in \mathbf{R}$ be a continuous function with $X_i$ sampled i.i.d. We say that $a_i(\theta)$ satisfies the ULLN property if (i) $\forall \theta, \mathbf{E}\left[a_i(\theta)^2\right] < \infty$; (ii) $a_i(\theta)$ is dominated by a function $A(X_i)$: $\forall \theta, |a_i(\theta)| \le A(X_i)$; and (iii) $\mathbf{E}[A(X_i)] < \infty$.*

**Proposition 1** (Consistency). *Suppose that (i) Assumption 2 holds, (ii) $\forall j \in [M]$, $\tilde{g}_{t,j}(\theta)$ satisfies Property 1, and (iii) $\forall (i,j) \in [M]^2$, $\left[\tilde{g}_t(\theta)\tilde{g}_t(\theta)^\top\right]_{i,j}$ satisfies Property 1; Then, $\widehat{\theta}_T^{(\pi)} \xrightarrow[T \to \infty]{p} \theta^*$.*

**Proposition 2** (Asymptotic normality). *Suppose that (i) $\widehat{\theta}_T^{(\pi)} \xrightarrow{p} \theta^*$; (ii) $\forall (i,j) \in [M] \times [D]$, $\left[\frac{\partial \tilde{g}_t}{\partial \theta}(\theta)\right]_{i,j}$ satisfies Property 1; (iii) $\exists \delta > 0$ such that $\mathbf{E}\left[\|\tilde{g}_i(\theta^*)\|^{2+\delta}\right] < \infty$, and (iv) (Selection ratio convergence) $\kappa_T^{(\pi)} \xrightarrow{p} k$ for some constant $k \in \Delta_\psi$. Then $\widehat{\theta}_T$ is asymptotically normal:*

$$\sqrt{T}(\widehat{\theta}_T^{(\pi)} - \theta^*) \xrightarrow{d} \mathcal{N}\left(0, \Sigma(\theta^*, k)\right),$$

*where $\Sigma(\theta^*, k)$ is a constant matrix that depends only on $\theta^*$ and $k$ (see Appendix A.2 for the complete expression). By Assumption 2(e) and the Delta method, $\widehat{\beta}_T$ is asymptotically normal:*

$$\sqrt{T}(\widehat{\beta}_T - \beta^*) \xrightarrow{d} \mathcal{N}\left(0, V(\theta^*, k)\right), \text{ where } V(\theta^*, k) = \nabla_\theta f_{tar}(\theta^*)^\top [\Sigma(\theta^*, k)] \nabla_\theta f_{tar}(\theta^*).$$

Proposition 2 shows that for a policy under which the selection ratio $\kappa_T$ converges in probability to a constant, the GMM estimator $\widehat{\theta}_T$ can be asymptotically normal. The specific order in which the data sources are queried does not affect asymptotic normality as long the selection ratio $\kappa_T^{(\pi)}$ converges.

**Input:** $T \in \mathbb{N}$, $e \in (0,1)$

1 $n = \lfloor Te \rfloor$;
2 Collect $n$ samples s.t. $\kappa_n = \mathrm{ctr}\,(\Delta_\psi)$;
3 $\widehat{\theta}_n = \mathrm{GMM}(H_n)$;
4 $\widehat{k} = \arg\min_{\kappa \in \Delta_\psi} V(\widehat{\theta}_n, \kappa)$;
5 Collect remaining samples such that
$\quad \kappa_T = \mathrm{proj}(\widehat{k}, \tilde{\Delta})$;
6 $\widehat{\theta}_T = \mathrm{GMM}(H_T)$;
**Output:** $f_{\mathrm{tar}}(\widehat{\theta}_T)$

(a) The Explore-then-Commit policy.

**Input:** $T \in \mathbb{N}$, $s \in (0,1)$

1 $\widehat{k} = \mathrm{ctr}\,(\Delta_\psi)$;
2 $J = \frac{1}{s}$;
3 **for** $j \in [1, 2, \ldots, J]$ **do**
4 $\quad t = \lfloor Tsj \rfloor$;
5 $\quad$ Collect $\lfloor Ts \rfloor$ samples s.t. $\kappa_t = \widehat{k}$;
6 $\quad \widehat{\theta}_t = \mathrm{GMM}(H_t, \widehat{W} = \widehat{W}_{\mathrm{valid}})$;
7 $\quad \widehat{k}_t = \arg\min_{\kappa \in \Delta_\psi} V(\widehat{\theta}_t, \kappa)$;
8 $\quad \widehat{k} = \mathrm{proj}(\widehat{k}_t, \tilde{\Delta}_{j+1}(\kappa_t))$;
9 **end**
10 $\widehat{\theta}_T = \mathrm{GMM}(H_T, \widehat{W} = \widehat{W}_{\mathrm{efficient}})$;
**Output:** $f_{\mathrm{tar}}(\widehat{\theta}_T)$

(b) The Explore-then-Greedy policy.

Figure 1: Algorithms for OMS-ETC and OMS-ETG.

A *fixed (or non-adaptive) policy*, denoted by $\pi_k$, has $\kappa_T = k$ for some constant $k \in \Delta_\psi$. Here, the collection decisions do not depend on the data and each data source is queried a fixed fraction of the time. By Proposition 2, $\widehat{\theta}_T^{(\pi_k)}$ is asymptotically normal. The *oracle policy*, denoted by $\pi^*$, is the fixed policy with the lowest asymptotic variance. Thus for $\pi^*$, we have $\kappa_T^{(\pi^*)} = \kappa^*$, where $\kappa^* = \arg\min_\kappa V(\theta^*, \kappa)$. We call $\kappa^*$ the *oracle selection ratio*. For the oracle policy, we have $\sqrt{T}(\widehat{\beta}_T^{(\pi^*)} - \beta) \xrightarrow{d} \mathcal{N}(0, V(\theta^*, \kappa^*))$. The following assumption ensures that $\kappa^*$ is unique and consequently the data collection decisions are meaningful.

**Assumption 3.** $\kappa^*$ *uniquely minimizes* $V(\theta^*, \kappa)$: $\forall \kappa \in \Delta_\psi$ *s.t.* $\kappa \neq \kappa^*$, $V(\theta^*, \kappa) > V(\theta^*, \kappa^*)$.

**Definition 2** (Asymptotic regret)**.** *The asymptotic regret captures how close the scaled asymptotic error of a given policy is to the oracle policy. We define the asymptotic regret of a policy $\pi$ as*

$$R_\infty(\pi) = \mathrm{AMSE}\left(\sqrt{T}\left(\widehat{\beta}_T^{(\pi)} - \beta^*\right)\right) - V(\theta^*, \kappa^*), \tag{1}$$

*where AMSE is the asymptotic MSE (i.e., the MSE of the limiting distribution).*

For any fixed policy $\pi_k$ such that $\kappa_T^{(\pi_k)} = k$ for some constant $k \neq \kappa^*$, we have $R_\infty(\pi_k) = [V(\theta^*, k) - V(\theta^*, \kappa^*)] > 0$ (by Assumption 3). This shows that a fixed policy suffers constant regret. This motivates the design of adaptive policies, where the regret asymptotically converges to zero.

### 4.1 Online Moment Selection via Explore-then-Commit (OMS-ETC)

OMS-ETC is inspired by the ETC strategy for multi-armed bandits (MABs) [27, Chapter 6]. Under OMS-ETC, we first explore by collecting a fixed number of samples for each choice. Then, we use these samples to estimate the oracle selection ratio $\kappa^*$. Finally, we commit to this ratio for the remaining time steps. We denote the OMS-ETC policy by $\pi_{\mathrm{ETC}}$ (Figure 1a).

The policy $\pi_{\mathrm{ETC}}$ is characterized by an exploration fraction $e \in (0,1)$. We first collect $Te$ samples by querying each data source equally so that $\kappa_{Te} = \mathrm{ctr}\,(\Delta_\psi)$. We then estimate $\widehat{\theta}_{Te}$ and obtain the plugin estimate of $\kappa^*$ as $\widehat{k} = \arg\min_{\kappa \in \Delta_\psi} V(\widehat{\theta}_{Te}, \kappa)$. The *feasible region* for $\kappa_T$ is defined as the set of values that $\kappa_T$ can take after we have devoted $Te$ samples to exploration and is given by $\tilde{\Delta} = \{e\kappa_{Te} + (1-e)\kappa : \kappa \in \Delta_\psi\}$ (proof in Appendix C). We collect the remaining $T(1-e)$ samples such that $\kappa_T$ is as close to $\widehat{k}$ as possible: $\kappa_T = \mathrm{proj}(\widehat{k}, \tilde{\Delta})$, where $\mathrm{proj}\left(k, \tilde{\Delta}\right) = \arg\min_{k' \in \tilde{\Delta}} \|k - k'\|$.

**Remark.** *The feasible region shrinks as $e$ increases because, as $e$ increases, the $T(1-e)$ samples that remain after exploration decrease thereby shrinking the possible values that $\kappa_T$ can take.*

**Theorem 1** (Regret of OMS-ETC)**.** *Suppose that (i) Conditions (i)-(iii) of Proposition 2 hold and (ii) Assumption 3 holds. Case (a): For a fixed $e \in (0,1)$, if $\kappa^* \in \tilde{\Delta}$, then the regret converges to zero:*

$R_\infty(\pi_{ETC}) = 0$. *If $\kappa^* \notin \tilde{\Delta}$, then $\pi_{ETC}$ suffers constant regret: $R_\infty(\pi_{ETC}) = r$ for some constant $r > 0$. Case (b): If $e$ depends on $T$ such that $e = o(1)$ and $Te \to \infty$ as $T \to \infty$ (e.g. $e = \frac{1}{\sqrt{T}}$), then $\forall \theta^* \in \Theta$, we have $R_\infty(\pi_{ETC}) = 0$.*

The theorem provides sufficient conditions for when the regret converges to zero. Case (a) of the theorem shows that if we explore for a fixed fraction of the horizon $T$, the regret will only converge to zero if $\kappa^*$ is inside the feasible region. Thus the regret will *not* converge to zero over the entire parameter space $\Theta$ as there would be certain parameter values for which $\kappa^*$ would be outside the feasible region. Case (b) shows that we *can* achieve zero asymptotic regret for every $\theta^* \in \Theta$ by setting $e$ such that it becomes asymptotically negligible ($e \in o(1)$). The main idea in the proof (see Appendix A.3) is to show that $\kappa_T \xrightarrow{p} \kappa^*$ and apply Proposition 2. In Case (b), the feasible region $\tilde{\Delta}$ asymptotically covers the entire simplex ($\tilde{\Delta} \xrightarrow{T \to \infty} \Delta_\psi$) and this is sufficient to show that $\kappa_T \xrightarrow{p} \kappa^*$.

### 4.2 Online Moment Selection via Explore-then-Greedy (OMS-ETG)

We extend OMS-ETC by periodically updating our estimate of $\kappa^*$ as we collect additional samples instead of committing to a value after exploration. The data is collected in batches. OMS-ETG (Figure 1b) is characterized by a batch fraction $s \in (0, 1)$. The algorithm runs for $J = \frac{1}{s}$ rounds and we collect $b = Ts$ samples in each round. In the first round, we explore and thus $\kappa_b = \text{ctr}(\Delta_\psi)$. After every round $j \in [J-1]$, we estimate $\widehat{\theta}_{bj}$ and the oracle selection ratio: $\widehat{k}_{bj} = \arg\min_{\kappa \in \Delta_\psi} V(\widehat{\theta}_{bj}, \kappa)$. The feasible region for round $j+1$ (the set of values of $\kappa_{b(j+1)}$ can take) is $\tilde{\Delta}_{j+1}(\kappa_{bj}) = \left\{ \frac{j\kappa_{bj}+\kappa}{j+1} : \kappa \in \Delta_\psi \right\}$ (proof in Appendix C). In round $(j+1)$, we (greedily) collect samples such that $\kappa_{b(j+1)}$ is as close to $\widehat{k}_{j+1}$ as possible: $\kappa_{b(j+1)} = \text{proj}\left(\widehat{k}_j, \tilde{\Delta}_{j+1}(\kappa_{bj})\right)$.

Theorem 2 states sufficient conditions for when OMS-ETG has zero regret. We first state a finite-sample tail bound for the two-step GMM estimator under adaptively collected (non-i.i.d.) data in Lemma 1 (which might be of independent interest) and use it to prove the theorem.

**Property 2** (Concentration). *Let $\tilde{a}_i(\theta) := \tilde{a}(X_i; \theta) \in \mathbf{R}$ with $X_i$ sampled i.i.d., $\tilde{a}_*(\theta) = \mathbf{E}[\tilde{a}(X_i; \theta)]$, $A(X_i, \theta) = \frac{\partial \tilde{a}(X_i; \theta)}{\partial \theta}$, $u_i(\eta) = \sup_{\theta, \theta' \in \Theta, \|\theta-\theta'\| \leq \eta} |\tilde{a}_i(\theta) - \tilde{a}_i(\theta')|$, and $u_*(\eta) = \mathbf{E}[u_i(\eta)]$. Let $L_1, \eta_0$, and $A_0$ be some positive constants. We say that $\tilde{a}_i(\theta)$ satisfies the Concentration property if (i) $\tilde{a}_*(\theta)$ is $L_1$-Lipschitz, (ii) $\forall \theta \in \Theta$, $[\tilde{a}_i(\theta) - \tilde{a}_*(\theta)]$ is sub-Exponential, (iii) $\mathbf{E}[\|A(X_i, \theta)\|] < A_0 < \infty$, and (iv) one of the following two conditions hold: (a) $\forall \eta \in (0, \eta_0)$, $[u_i(\eta) - u_*(\eta)]$ is sub-Exponential, or (b) $\sup_{\theta \in \Theta} \|A(X_i, \theta)\|$ is sub-Exponential.*

**Remark.** *Property 2 is used to derive a uniform law (see Lemma 4) that is used to prove Lemma 1. Property 2(iv) might be hard to check but (iv)(a) is satisfied for bounded function classes, i.e., when $\|\tilde{a}_i\|_\infty < A < \infty$ (see Proposition 9) and (iv)(b) for linear function classes with sub-Exponential data (see Proposition 10). For the linear IV model in Example 1, Property 2(iv) would hold if $Z_i^2$ is sub-Exponential (e.g., when $Z_i$ is sub-Gaussian).*

**Lemma 1** (GMM concentration inequality). *Let $\lambda_*, C_0, \eta_1, \eta_2$, and $\delta_0$ be some positive constants. Suppose that (i) Assumption 2 holds; (ii) $\forall j$, $\tilde{g}_{i,j}(\theta)$ satisfies Property 2; (iii) The spectral norm of the GMM weight matrix $\widehat{W}$ is upper bounded with high probability: $\forall \delta \in (0, C_0)$, $\mathbf{P}\left(\|\widehat{W}\| \leq \lambda_*\right) \geq 1 - \frac{1}{\delta^D} \exp\left\{-\mathcal{O}\left(T\delta^2\right)\right\}$ (see Remark 1); (iv) (Local strict convexity) $\forall \theta \in N_{\eta_1}(\theta^*)$, $\mathbf{P}\left(\left\|\frac{\partial^2 \bar{Q}}{\partial \theta^2}(\theta)^{-1}\right\| \leq h\right) = 1$ ($\bar{Q}(\theta)$ is defined in Assumption 2a); (v) (Strict minimization) $\forall \theta \in N_{\eta_2}(\theta^*)$, there is a unique minimizer $\kappa(\theta) = \arg\min_\kappa V(\theta, \kappa)$ s.t. $V(\theta, \kappa) - V(\theta, \kappa(\theta)) \leq c\delta^2 \implies \|\kappa - \kappa(\theta)\| \leq \delta$; and (vi) $\sup_\kappa |V(\theta, \kappa) - V(\theta', \kappa)| \leq L\|\theta - \theta'\|$. Then, for $\widehat{k}_t = \arg\min_{\kappa \in \Delta_\psi} V(\widehat{\theta}_T^{(\pi)}, \kappa)$, any policy $\pi$, and $\forall \delta \in (0, \delta_0)$,*

$$\mathbf{P}\left(\left\|\widehat{\theta}_T^{(\pi)} - \theta^*\right\| > \delta\right) < \frac{1}{\delta^{2D}} \exp\left\{-\mathcal{O}\left(T\delta^4\right)\right\} \ and \ \mathbf{P}\left(\left\|\widehat{k}_T - \kappa^*\right\| > \delta\right) < \frac{1}{\delta^{4D}} \exp\left\{-\mathcal{O}\left(T\delta^8\right)\right\}.$$

*Better rates for $\widehat{k}_T$ are applicable under additional restrictions on $\theta^*$ (see Lemma 6).*

To derive the tail bound for $\widehat{\theta}_T$, we first prove that the minimized empirical GMM objective is close to $\bar{Q}(\theta^*)$ with high probability (w.h.p.) (using Conditions (i)-(iii)). Then we show that $\widehat{\theta}_T$ is close to

$\theta^*$ w.h.p. (using Condition (iv)). Next, we use the inequality for $\widehat{\theta}_T$ to derive the tail bound for $\widehat{k}$. For this, we show that $V(\theta^*, \widehat{k})$ is close to $V(\theta^*, \kappa^*)$ (using Condition (vi)) and then show that $\widehat{k}$ is close to $\kappa^*$ w.h.p. (using Condition (v)).

**Theorem 2** (Regret of OMS-ETG). *Suppose that Conditions (i)-(iv) of Proposition 2 hold. Let $\tilde{\Delta}(s) = \{s\kappa_b + (1-s)\kappa : \kappa \in \Delta_\psi\}$. Case (a): For a fixed $s \in (0,1)$, if the oracle selection ratio $\kappa^* \in \tilde{\Delta}(s)$, then the regret converges to zero: $R_\infty(\pi_{ETG}) = 0$. If $\kappa^* \notin \tilde{\Delta}(s)$, then $R_\infty(\pi_{ETG}) > 0$ (non-zero regret). Case (b): Now also suppose that the conditions for Lemma 1 hold. If $s = CT^{\eta-1}$ for some constant $C$ and any $\eta \in [0,1)$, then $\forall \theta^* \in \Theta, \; R_\infty(\pi_{ETG}) = 0$.*

Similar to OMS-ETC, Case (a) of the theorem shows that if the batch size is a constant fixed fraction, some values of $\kappa^*$ will be outside the feasible region and thus we do not get zero regret over the entire parameter space. Case (b) shows that to get zero asymptotic regret for every $\theta^* \in \Theta$, $s$ must depend on $T$ and be asymptotically negligible. But unlike OMS-ETC, the estimate of $\kappa^*$ is updated after every round. The batch fraction $s$ can be as small as $CT^{-1}$ allowing the agent to collect a constant number of samples in each batch.

We prove the theorem by showing that $\kappa_T \xrightarrow{p} \kappa^*$ and applying Proposition 2. To do so for Case (b), we show that the estimated oracle ratio after *every* round is close to $\kappa^*$, i.e., $\forall \epsilon > 0, \; \mathbf{P}\left(\forall j \in [J-1], \widehat{k}_{bj} \in N_\epsilon(\kappa^*)\right) \xrightarrow[T \to \infty]{} 1$ (by using Lemma 1). Since we move as close as possible to $\widehat{k}_{bj}$ after every round, this ensures that $\forall \epsilon > 0, \; \mathbf{P}\left(\kappa_T \in N_\epsilon(\kappa^*)\right) \to 1$ (and thus $\kappa_T \xrightarrow{p} \kappa^*$).

Both OMS-ETC and OMS-ETG are asymptotically equivalent as they can achieve zero regret for every $\theta^* \in \Theta$. But our experiments show that OMS-ETG outperforms OMS-ETC (in terms of regret) for small sample sizes. This may be because with OMS-ETG, the estimate of $\kappa^*$ keeps improving as more samples are collected instead of being fixed after exploration. This suggests that better estimates of $\kappa^*$ may lead to higher-order reductions in MSE.

**Remark 1** (Weight matrix $\widehat{W}$). *For OMS-ETG, the GMM weight matrix $\widehat{W}$ needs to satisfy Condition (iii) of Lemma 1 till round $(J-1)$. We denote this matrix by $\widehat{W}_{valid}$ in Figure 1b. For the final step of OMS-ETG, we use the standard efficient two-step GMM weight matrix (denoted by $\widehat{W}_{efficient}$ in Figure 1b) and thus the final estimate of $\theta^*$ is still asymptotically efficient. Condition (iii) of Lemma 1 would hold for the efficient two-step GMM weight matrix (i.e., for $\widehat{W} := \left[\widehat{\Omega}_T(\widehat{\theta}_T^{(os)})\right]^{-1}$) if $\forall(j,k), [\tilde{g}_{i,j}(\theta)\tilde{g}_{i,k}(\theta)]$ satisfies Property 2 (see Lemma 5 for proof). This would hold if the moments $\tilde{g}_{i,j}(\theta)$ are uniformly bounded. Condition (iii) can also be satisfied with a regularized weight matrix: $\widehat{W} := \left[\widehat{\Omega}_T(\widehat{\theta}_T^{(os)}) + \lambda_W I\right]^{-1}$ for some $\lambda_W > 0$ as this ensures that $\|\widehat{W}\| \leq \lambda_W^{-1}$.*

**Additional exploration.** MAB algorithms usually require additional exploration to perform well (e.g., $\epsilon$-greedy and upper confidence bound strategies [27, Chapter 7]). However, we empirically noticed that additional exploration hurts performance in our setup. This might be because unlike typical bandit setups where pulling one arm does not improve the estimates of another arm, querying any data source can improve the estimates of the model parameter $\theta^*$ in our setup.

## 5  Incorporating a Cost Structure

In many real-world settings, the agent has a budget constraint and must pay a different cost to query each data source. We adapt OMS-ETC and OMS-ETG to this setting where a cost structure is associated with the data sources in $\psi$. We prove that these policies still have zero asymptotic regret for every $\theta^* \in \Theta$. Let $(\psi_i)_{i \in [|\psi|]}$ be an indexed family. We denote the (known) budget by $B \in \mathbb{N}$ and by $c \in \mathbf{R}_{>0}^{|\psi|}$, a cost vector such that $c_i$ is the cost of querying data source $\psi_i$. Due to the cost structure, the horizon $T$ is a random variable dependent on $\pi$ with $T = \left\lfloor \frac{B}{\kappa_T^\top c} \right\rfloor$. The setting in Section 4 is a special case of this formulation when $\forall i, c_i = 1$ and $B = T$. We defer proofs to Appendix B.

For a fixed policy $\pi_k$, we have $\kappa_T^{(\pi_k)} = k$, for some constant $k \in \Delta_\psi$. By Proposition 2, as $B \to \infty$, we have $\sqrt{B}\left(\widehat{\beta}_T^{(\pi_k)} - \beta^*\right) \xrightarrow{d} \mathcal{N}\left(0, V(\theta^*, k)(k^\top c)\right)$. Here we scale by $\sqrt{B}$ instead of

$\sqrt{T}$ to make comparisons across policies meaningful. The *oracle selection ratio* is now defined as $\kappa^* = \arg\min_{\kappa \in \Delta_\psi} \left[ V(\theta^*, \kappa) \left( \kappa^\top c \right) \right]$ and the *asymptotic regret* of policy $\pi$ now is

$$R_\infty(\pi) = \text{AMSE}\left( \sqrt{B} \left( \widehat{\beta}_T^{(\pi)} - \beta^* \right) \right) - V(\theta^*, \kappa^*) \left( (\kappa^*)^\top c \right).$$

OMS-ETC-CS (*OMS-ETC with cost structure*) is an adaptation of OMS-ETC for this setting. We use $Be$ budget to explore and estimate $\kappa^*$ by $\widehat{k} = \arg\min_{\kappa \in \Delta_\psi} \left[ V(\widehat{\theta}_{T_e}, \kappa) \left( \kappa^\top c \right) \right]$, where $T_e = \left\lfloor \frac{eB}{\kappa_{T_e}^\top c} \right\rfloor$ and $\kappa_{T_e} = \text{ctr}(\Delta_\psi)$. Exploration strategies that utilize the cost structure can also be used (e.g., evenly dividing the budget across data sources while exploring). With the remaining budget, we collect samples such that $\kappa_T = \text{proj}\left( \widehat{k}, \tilde{\Delta} \right)$, where $\tilde{\Delta}$ is the feasibility region (expression given in Appendix C). The following proposition shows that OMS-ETC-CS can achieve zero regret.

**Proposition 3** (Regret of OMS-ETC-CS). *Suppose that the conditions of Theorem 1 hold. If $e = o(1)$ such that $Be \to \infty$ as $B \to \infty$, then $\forall \theta^* \in \Theta$, $R_\infty(\pi_{ETC\text{-}CS}) = 0$.*

We propose two ways of adapting OMS-ETG to this setting: (i) OMS-ETG-FS (*OMS-ETG with fixed samples*) where we collect a fixed number of samples in every round and (ii) OMS-ETG-FB (*OMS-ETG with fixed budget*) where we spend a fixed fraction of the budget in every round.

Let $c_{\max} = \max_{i \in [|\psi|]} c_i$ and $c_{\min} = \min_{i \in [|\psi|]} c_i$. In OMS-ETC-FS (Figure 6a), we collect $b = \left\lfloor \frac{Bs}{c_{\max}} \right\rfloor$ samples in every round except the last one (the last batch can be smaller as we may not have enough budget left for a full batch). The number of rounds $J$ is random since, in every round, depending on what we collect, we use up a different amount of the budget. Like OMS-ETG, after each round, we estimate $\kappa^*$ and greedily collect samples to get as close to it as possible.

In OMS-ETG-FB (Figure 6b), we spend $Bs$ budget in each round. Thus the number of rounds $J$ is fixed with $J = \frac{1}{s}$ but the number of samples collected per round is now random. Like OMS-ETG, after each round, we estimate $\kappa^*$ and collect samples to get as close to the estimate as possible. The next two Propositions show that both OMS-ETG-FS and OMS-ETC-FB have zero asymptotic regret.

**Proposition 4** (Regret of OMS-ETG-FS). *Suppose that the conditions of Theorem 2b hold. If $s = B^{\eta-1}$ for any $\eta \in [0, 1)$, then $\forall \theta^* \in \Theta$, $R_\infty(\pi_{ETG\text{-}FS}) = 0$.*

**Proposition 5** (Regret of OMS-ETG-FB). *Suppose that the conditions of Theorem 2b hold. If $s = B^{\eta-1}$ for any $\eta \in [0, 1)$, then $\forall \theta^* \in \Theta$, $R_\infty(\pi_{ETG\text{-}FB}) = 0$.*

## 6 Experiments

### 6.1 Synthetic data

We validate our methods on synthetic data generated from known causal graphs (see Appendix D for parameter values and moment conditions used). In our experiments (including the ones in Section 6.2), for OMS-ETG, we use the regularized GMM weight matrix with $\lambda_W := 0.01$ (see Remark 1). The regret is only minimally affected with an unregularized matrix ($\lambda_W := 0$) despite theoretical guarantees only holding for the regularized case (the maximum change in regret was $1.98\%$) and thus our conclusions do not change. We first simulate data from a linear IV graph (Figure 2a) and compare the performance of different polices (Figure 3a). We use $\psi = \{\{Z, X\}, \{Z, Y\}\}$ and assume that both sources cost the same. We set parameter values such that $\kappa^* \approx [0.36, 0.64]^\top$. We compare policies based on *relative regret (RR)* with respect to the oracle policy:

$$\text{Relative regret} = \text{RR}(\pi) = \frac{\text{MSE}^{(\pi)} - \text{MSE}^{(\text{oracle})}}{\text{MSE}^{(\text{oracle})}} \times 100\%.$$

The MSE values are computed across $12{,}000$ runs. The label *etc_{x}* in the plot refers to OMS-ETC with exploration fraction $e = x$ (e.g., *etc_0.1* means $e = 10\%$). Similarly, *etg_{x}* refers to OMS-ETG with $s = x$. We see that as the horizon increases, the RR of all policies converges to zero. This supports the claim that both OMS-ETC and OMS-ETG have zero asymptotic regret. However, when the horizon is small ($T = 300$), both *etc_0.1* and *etc_0.2* perform poorly due to insufficient exploration. In contrast, OMS-ETG has close to zero RR even for small horizons which shows that repeatedly improving the parameter estimates can lead to faster convergence of regret.

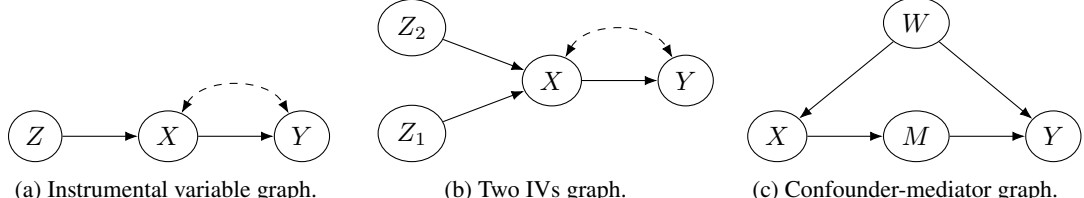

(a) Instrumental variable graph.   (b) Two IVs graph.   (c) Confounder-mediator graph.

Figure 2: Examples of causal models—with treatment $X$ and outcome $Y$—where the ATE can be identified by different data sources returning different subsets of variables.

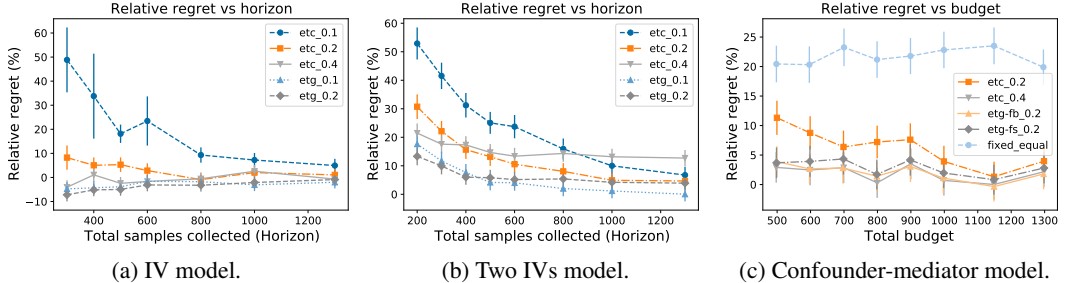

(a) IV model.   (b) Two IVs model.   (c) Confounder-mediator model.

Figure 3: Relative regret (RR) across horizons/budgets for different policies (error bars denote $95\%$ CIs). (a) RR for the IV model (Figure 2a) with $\psi = \{\{Z, X\}, \{Z, Y\}\}$. (b) RR for the two IVs model (Figure 2b) with $\psi = \{\{X, Y, Z_1\}, \{X, Y, Z_2\}\}$. (c) RR for the confounder-mediator model (Figure 2c) with $\psi = \{\{X, Y, W\}, \{X, Y, M\}\}$ and a cost structure.

Next, we simulate data from a linear graph with two IVs (Figure 2b). Here $\psi = \{\{X, Y, Z_1\}, \{X, Y, Z_2\}\}$ and both choices cost the same. We set the parameters such that $\kappa^* = [0, 1]^\top$ ($\kappa^*$ is on the corner of the simplex). We compare the RR across various horizons (Figure 3b). We see the OMS-ETC performs worse than OMS-ETG for small horizons. One difference from the previous case (Figure 3a) is that *etc_0.4* performs poorly even for large horizons. This is because after using $40\%$ of the samples for exploration, the feasibility region is not large enough to get close to the corner of the simplex. This demonstrates another benefit of OMS-ETG over OMS-ETC: OMS-ETG can achieve close to zero regret in finite samples when the oracle ratio $\kappa^*$ is either on the boundary or in the interior of the simplex.

Finally, we simulate data from a linear confounder-mediator graph (Figure 2c). Here, both the backdoor (using $\{X, Y, W\}$) and frontdoor (using $\{X, Y, M\}$) adjustments are applicable [31, Section 3.3]. We use $\psi = \{\{X, Y, W\}, \{X, Y, M\}\}$ with cost structure $c = [1.8, 1]^\top$ (confounders $W$ cost more than the mediators $M$). We set the parameters such that $\kappa^* \approx [0.15, 0.85]^\top$. We see similar conclusions as the previous cases. OMS-ETC with low exploration performs poorly but converges for large horizons. Both OMS-ETG variants—OMS-ETG-FS and OMS-ETG-FB—have close to zero RR for all horizons. We see no significant difference between the regret of OMS-ETG-FS and OMS-ETG-FB. The policy *fixed_equal* is a fixed policy that collects an equal fraction of both subsets. Its RR does not converge and is substantially higher than the oracle ($\approx 20\%$). This demonstrates that adaptive policies can lead to significant gains in MSE over fixed policies and that our methods remain applicable even with an associated cost structure on the data sources.

## 6.2 Semi-synthetic data

**IHDP.** Hill [19] constructed a dataset based on the Infant Health and Development Program (IHDP). The data [10] is from a randomized experiment studying the effect of home visits by a trained provider on future cognitive test scores of children. Following Hill [19], we create an observational dataset by removing a non-random subset of the data. The treatment $X$ is binary. The dataset contains pre-treatment covariates which are measurements on the mother and the child. For simplicity, we only use two covariates: birth weight (continuous) ($W_1$) and whether the mother smoked (binary) ($W_2$). For each sample of the generated semi-synthetic data, $(X, W_1, W_2)$ are sampled uniformly at random from the real data. The outcome $Y$ (continuous) is simulated: $Y := \alpha_1 W_1 + \alpha_2 W_2 + \beta X + \epsilon_y$,

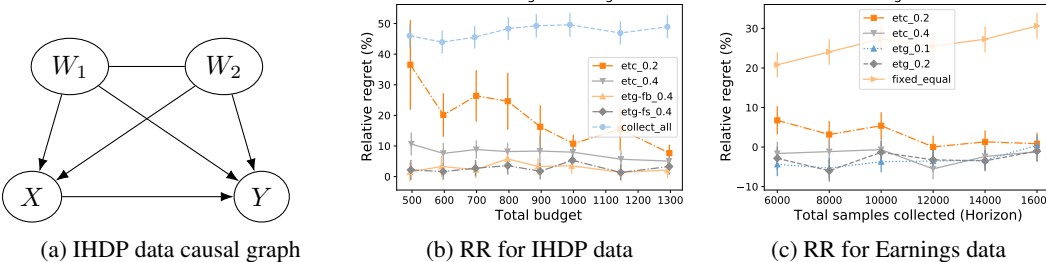

(a) IHDP data causal graph  (b) RR for IHDP data  (c) RR for Earnings data

Figure 4: Relative regret (RR) on semi-synthetic data (error bars denote $95\%$ CIs). For both IHDP (b) and Earnings data (c), adaptive policies converge to zero RR but fixed policies (*collect_all*, *fixed_equal*) suffer constant regret. OMS-ETG outperforms OMS-ETC for small horizons.

where $\epsilon_y \sim \mathcal{N}(0, \sigma_y^2)$, $\alpha_1, \alpha_2, \beta \in \mathbf{R}$, and $\sigma_y^2 \in \mathbf{R}^+$ (see Figure 4a). Here $\alpha_1, \alpha_2, \beta$, and $\sigma_y$ are model parameters with $\beta$ being the ATE (target parameter).

For this experiment, we use $\psi = \{\{X, Y, W_1\}, \{X, Y, W_2\}, \{X, Y, W_1, W_2\}\}$ with cost structure $c = [1, 3, 3.5]^\top$. Thus, at each step, the agent can collect either one of the covariates or both of them, and each choice has a distinct cost. Setting model parameters such that $\kappa^* \approx [0.59, 0, 0.41]^\top$, we compare performance across policies for various total budgets (Figure 4b). The policy *collect_all* is a fixed policy that collects $\{X, Y, W_1, W_2\}$ at every step. This policy has higher RR ($\approx 50\%$ higher MSE than the oracle) for all budgets demonstrating that *collecting all covariates for every sample can be sub-optimal*. The policy *etc_0.2* does poorly with a small budget whereas *etc_0.4* has close to zero RR. Both OMS-ETG-FB and OMS-ETG-FS have close to zero RR for all horizons.

**The Vietnam draft and future earnings.** Angrist [2] computed the effect of veteran status on future earnings from the Vietnam draft lottery data [3] using an IV (Figure 2a). The IV $Z$ (binary) indicates whether an individual was eligible for the draft based on a random lottery. The treatment $X$ (binary) indicates whether they actually served. The outcome $Y$ (continuous) represents their future earnings. The IV removes bias caused by certain types of men being more likely to serve. In this dataset, $\{Z, X\}$ and $\{Z, Y\}$ were collected using different data sources (thus $\{Z, X, Y\}$ are not observed simultaneously), which suits our framework. We construct a semi-synthetic dataset that closely matches the real data so that we know the ground-truth causal effect (needed to compute the MSE). For each instance, we sample $Z$ uniformly at random from the empirical distribution. $X$ is sampled from the Bernoulli distribution $\widehat{\mathbf{P}}(X|Z)$ with conditional probabilities given by the empirical distribution (values taken from [2, Table 2]). We generate the outcome as $Y := \beta X + \gamma + \epsilon$, where $\epsilon \sim \mathcal{N}(0, \sigma_\epsilon^2)$ and $\epsilon \not\perp X$. The parameters $\beta, \gamma$, and $\sigma_\epsilon^2$ are set such that the distribution of $(Z, Y)$ is close to the real data (see Appendix D.5 for details). We compare the RR of our policies on this dataset (Figure 4c). Most adaptive policies converge to near zero RR as the horizon gets large. OMS-ETC does poorly with low exploration while OMS-ETG policies have significantly lower RR for smaller horizons. By contrast, *fixed_equal* (the fixed policy that queries both data sources equally) suffers constant regret and has $\approx 25\%$ higher regret than the oracle even for large horizons. This demonstrates that adaptive policies can lead to substantial MSE gains in a real-world setting.

# 7 Conclusion

This paper takes some initial strides towards endogenizing decisions about which variables to solicit into the modeling process. Addressing the problem of deciding, sequentially, which data sources to query in order to efficiently estimate a parameter, we developed the online moment selection (OMS) framework and two instantiations: OMS-ETC and OMS-ETG. We prove that over the entire parameter space, adaptive data collection with either method can provide substantial MSE gains. While our work focuses on ATE estimation, our framework is more broadly applicable to any parameter identified by moment conditions. In future work, we hope to apply our framework to more general prediction problems, addressing practical considerations including high-dimensional data and complex model classes (e.g., neural networks). Moreover, in real-world settings, common assumptions like ignorability rarely hold. We hope to extend our framework to overcome issues such as model misspecification, or to overcome biases present in some, but not all data sources.

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
