# A  Omitted Proofs for Section 4

## A.1  Proof of Proposition 1 (Consistency)

**Proposition 6** (MDS LLN [16, Example 7.11]). *Let $\bar{Y}_T$ be the sample mean from a martingale difference sequence (MDS), $\bar{Y}_T = \frac{1}{T} \sum_{t=1}^{T} Y_i$, with $\mathbf{E}\left[|Y_t|^r\right] < \infty$ for some $r > 1$. Then $\bar{Y}_T \xrightarrow{p} 0$.*

**Lemma 2** (Uniform convergence). *Let $a_i(\theta) := S_i \tilde{a}(\theta, X_i)$ be a real-valued function where $S_i \in \{0, 1\}$ is $H_{i-1}$-measurable and $X_i$ are i.i.d. Suppose that (i) $\Theta$ is compact and (ii) $\tilde{a}(\theta, X_i)$ satisfies Property 1. Then*

$$\sup_{\theta \in \Theta} \left| \frac{1}{T} \sum_{i=1}^{T} [a_i(\theta) - S_i a_*(\theta)] \right| \xrightarrow{p} 0,$$

*where $a_*(\theta) = \mathbf{E}[\tilde{a}(\theta; X_i)]$.*

*Proof.* We follow a standard uniform law of large numbers proof (e.g. Tauchen [39, Lemma 1]) and modify it to work for dependent data. The key modification is replacing the law of large numbers (LLN) in that proof with a MDS LLN.

Let $(\theta_1, \theta_2, \ldots, \theta_K)$ be a minimal $\delta$-cover of $\Theta$ and $N_\delta(\theta_k)$ denote the $\delta$-ball around $\theta_k$. By compactness of $\Theta$, $K$ is finite. For $k \in [K]$ and $\theta \in N_\delta(\theta_k)$, we have

$$\left| \frac{1}{T} \sum_{i=1}^{T} [a_i(\theta) - S_i a_*(\theta)] \right|$$

$$= \left| \frac{1}{T} \sum_{i=1}^{T} [a_i(\theta) - a_i(\theta_k) + a_i(\theta_k) - S_i a_*(\theta_k) + S_i a_*(\theta_k) - S_i a_*(\theta)] \right|$$

$$\leq \frac{1}{T} \sum_{i=1}^{T} |a_i(\theta) - a_i(\theta_k)| + \left| \frac{1}{T} \sum_{i=1}^{T} [a_i(\theta_k) - S_i a_*(\theta_k)] \right| + \frac{1}{T} \sum_{i=1}^{T} |S_i a_*(\theta_k) - S_i a_*(\theta)|$$

$$= \frac{1}{T} \sum_{i=1}^{T} |S_i (\tilde{a}(\theta; X_i) - \tilde{a}(\theta_k; X_i))| + \left| \frac{1}{T} \sum_{i=1}^{T} [a_i(\theta_k) - S_i a_*(\theta_k)] \right| + \frac{1}{T} \sum_{i=1}^{T} |S_i (a_*(\theta_k) - a_*(\theta))|$$

$$\leq \frac{1}{T} \sum_{i=1}^{T} |\tilde{a}(\theta; X_i) - \tilde{a}(\theta_k; X_i)| + \left| \frac{1}{T} \sum_{i=1}^{T} [a_i(\theta_k) - S_i a_*(\theta_k)] \right| + |a_*(\theta_k) - a_*(\theta)|.$$

We now show that each of the three terms on the RHS above is small. In the third term, by continuity of $a_*(\theta)$, $\forall \epsilon > 0, \exists \delta > 0$ s.t. $|a_*(\theta_k) - a_*(\theta)| < \epsilon$.

In the second term, $[a_i(\theta_k) - S_i a_*(\theta_k; S_i)]$ is a MDS. By Property 1(i) and Proposition 6, we have $\left| \frac{1}{T} \sum_{i=1}^{T} [a_i(\theta_k) - S_i a_*(\theta_k)] \right| \xrightarrow{p} 0$.

Next, we examine first term on the RHS. Let $u_i(\delta) = \sup_{\theta, \theta' \in \Theta, \|\theta - \theta'\| \leq \delta} |\tilde{a}(\theta, X_i) - \tilde{a}(\theta', X_i)|$. By continuity of $\tilde{a}(\theta, X_i)$, compactness of $\Theta$, and the Heine-Cantor theorem, $\tilde{a}(\theta, X_i)$ is uniformly continuous in $\theta$. This ensures that $u_i(\delta)$ is continuous in $\delta$ and thus $u_i(\delta) \downarrow 0$ as $\delta \downarrow 0$. Since $u_i(\delta) \leq 2A(X_i)$ (by Property 1(iii)), using dominated convergence, we have $\mathbf{E}[u_i(\delta)] \downarrow 0$ as $\delta \downarrow 0$. Therefore, $\forall \epsilon > 0, \exists \delta > 0$ s.t. $\mathbf{E}[u_i(\delta)] < \epsilon$. Thus we can write the first term as

$$\frac{1}{T} \sum_{i=1}^{T} |\tilde{a}(\theta; X_i) - \tilde{a}_i(\theta_k; X_i)| \leq \frac{1}{T} \sum_{i=1}^{T} u_i(\delta)$$

$$= \frac{1}{T} \sum_{i=1}^{T} u_i(\delta) - \mathbf{E}[u_i(\delta)] + \mathbf{E}[u_i(\delta)]$$

$$\leq \frac{1}{T} \sum_{i=1}^{T} u_i(\delta) - \mathbf{E}[u_i(\delta)] + \epsilon$$

$$\overset{(a)}{=} o_p(1) + \epsilon,$$

where (a) follows by the weak law of large numbers which applies because $E[u_i(\delta)] \leq \mathbf{E}[A(X_i)] < \infty$ (by Property 1(iii)). $\square$

**Proposition** (Consistency). *Suppose that (i) Assumption 2 holds, (ii) $\forall j \in [M]$, $\tilde{g}_{t,j}(\theta)$ satisfies Property 1, and (iii) $\forall (i,j) \in [M]^2$, $[\tilde{g}_t(\theta)\tilde{g}_t(\theta)^\top]_{i,j}$ satisfies Property 1; Then, for any policy $\pi$, $\widehat{\theta}_T^{(\pi)} \xrightarrow[T\to\infty]{p} \theta^*$.*

*Proof.* We begin by defining the empirical and population analogues of the two-step GMM objective for a given policy $\pi$:

$$\text{Empirical objective: } \widehat{Q}_T^{(\pi)}(\theta) = \left[\frac{1}{T}\sum_{t=1}^{T} g_t(\theta)\right]^\top \widehat{W} \left[\frac{1}{T}\sum_{t=1}^{T} g_t(\theta)\right],$$

$$\text{Population objective: } \bar{Q}_T^{(\pi)}(\theta) = \left[\frac{1}{T}\sum_{t=1}^{T} \mathbf{E}\left[g_t(\theta)|H_{t-1}\right]\right]^\top W \left[\frac{1}{T}\sum_{t=1}^{T} \mathbf{E}\left[g_t(\theta)|H_{t-1}\right]\right]$$

$$= \left[\left(\frac{1}{T}\sum_{t=1}^{T} m(s_t)\right) \otimes g_*(\theta)\right]^\top W \left[\left(\frac{1}{T}\sum_{t=1}^{T} m(s_t)\right) \otimes g_*(\theta)\right]$$

$$= \left[m_T \otimes g_*(\theta)\right]^\top W \left[m_T \otimes g_*(\theta)\right],$$

where $g_*(\theta) = \mathbf{E}\left[\tilde{g}_i(\theta)\right]$ and $m_T = \frac{1}{T}\sum_{t=1}^{T} m(s_t)$. We have $\widehat{W} = \left[\widehat{\Omega}_T(\widehat{\theta}_T^{(\text{os})})\right]^{-1}$, where $\widehat{\theta}_T^{(\text{os})}$ is the one-step GMM estimate and

$$\widehat{\Omega}_T(\theta) = \frac{1}{T}\sum_{t=1}^{T}\left[g_t(\theta)g_t(\theta)^\top\right]$$

$$= \frac{1}{T}\sum_{t=1}^{T}\left(\left[m(s_t)m(s_t)^\top\right] \otimes \left[\tilde{g}_t(\theta)\tilde{g}_t(\theta)^\top\right]\right).$$

Furthermore, we have $W = [m_\Omega(\kappa_T) \otimes \Omega(\theta^*)]^{-1}$, where

$$m_\Omega(\kappa_T) = \sum_{t=1}^{T}\left(m(s_t)m(s_t)^\top\right),$$

$$\Omega(\theta) = \mathbf{E}\left[\tilde{g}_t(\theta)\tilde{g}_t(\theta)^\top\right].$$

The two-step GMM estimator is obtained by minimizing the empirical objective: $\widehat{\theta}_T = \arg\min_{\theta\in\Theta}\widehat{Q}_T^{(\pi)}(\theta)$. At the true parameter $\theta^*$, $\bar{Q}_T^{(\pi)}(\theta^*) = 0$ and by Assumption 2(a), $\theta^*$ uniquely minimizes $\bar{Q}_T^{(\pi)}(\theta)$. By Newey and McFadden [30, Theorem 2.1], $\sup_{\theta\in\Theta}\left|\widehat{Q}_T^{(\pi)}(\theta) - \bar{Q}_T^{(\pi)}(\theta)\right| \xrightarrow{p} 0 \implies \widehat{\theta}_T \xrightarrow{p} \theta^*$.

**Uniform convergence of $\widehat{Q}_T^{(\pi)}(\theta)$.** We now prove that $\sup_{\theta\in\Theta}\left|\widehat{Q}_T^{(\pi)}(\theta) - \bar{Q}_T^{(\pi)}(\theta)\right| \xrightarrow{p} 0$. Following the proof of Newey and McFadden [30, Theorem 2.6], we have

$$\left|\widehat{Q}_T^{(\pi)}(\theta) - \bar{Q}_T^{(\pi)}(\theta)\right|$$

$$\leq \left\|\frac{1}{T}\sum_{t=1}^{T}\left[g_t(\theta) - m(s_t) \otimes g_*(\theta)\right]\right\|^2 \left\|\widehat{W}\right\|^2 + 2\left\|g_*(\theta)\right\|\left\|\frac{1}{T}\sum_{t=1}^{T}\left[g_t(\theta) - m(s_t) \otimes g_*(\theta)\right]\right\|\left\|\widehat{W}\right\| +$$

$$\left\|g_*(\theta)\right\|^2\left\|\widehat{W} - W\right\|.$$

$$(2)$$

We first prove that $\left\|\widehat{W} - W\right\| \xrightarrow{p} 0$. Due to Condition (iii) of the theorem, we can apply Lemma 2 to get

$$\forall\,(i,j) \in [M]^2,\ \forall\,\epsilon > 0,\ \mathbf{P}\left(\sup_{\theta \in \Theta}\left|\widehat{\Omega}_T(\theta)_{i,j} - [m_\Omega(\kappa_T) \otimes \Omega(\theta)]\right| > \epsilon\right) \to 0,$$

$$\therefore\ \forall\,(i,j) \in [M]^2,\ \forall\,\epsilon > 0,\ \mathbf{P}\left(\left|\widehat{\Omega}_T(\widehat{\theta}_T^{(os)})_{i,j} - \left[m_\Omega(\kappa_T) \otimes \Omega(\widehat{\theta}_T^{(os)})\right]\right| > \epsilon\right) \to 0,$$

$$\therefore\ \forall\,(i,j) \in [M]^2,\ \forall\,\epsilon > 0,\ \mathbf{P}\left(\left|\widehat{\Omega}_T(\widehat{\theta}_T^{(os)})_{i,j} - [m_\Omega(\kappa_T) \otimes \Omega(\theta^*)]\right| > \epsilon\right) \xrightarrow{(a)} 0,$$

$$\therefore\ \forall\,(i,j) \in [M]^2,\ \forall\,\epsilon > 0,\ \mathbf{P}\left(\left|\widehat{W}_{i,j} - W_{i,j}\right| > \epsilon\right) \xrightarrow{(b)} 0,$$

$$\therefore\ \left\|\widehat{W} - W\right\| \xrightarrow{p} 0,$$

where (a) follows because $\widehat{\theta}_T^{(os)} \xrightarrow{p} \theta^*$ (by Proposition 1) and (b) by the continuous mapping theorem. Therefore, we have

$$\left\|\widehat{W}\right\| \leq \|W\| + o_p(1)$$

$$\leq \underbrace{\limsup_{T \to \infty}\left\|[m_\Omega(\kappa_T) \otimes \Omega(\theta^*)]^{-1}\right\|}_{:=\lambda_0} + o_p(1).$$

Substituting these results in Eq. 2, we get

$$\left|\widehat{Q}_T^{(\pi)}(\theta) - \bar{Q}_T^{(\pi)}(\theta)\right| \leq \left\|\frac{1}{T}\sum_{t=1}^T[g_t(\theta) - m(s_t) \otimes g_*(\theta)]\right\|^2 \lambda_0^2 + 2\|g_*(\theta)\|\left\|\frac{1}{T}\sum_{t=1}^T[g_t(\theta) - m(s_t) \otimes g_*(\theta)]\right\|\lambda_0 + o_p(1).$$

Thus, to show uniform convergence of $\widehat{Q}_T^{(\pi)}(\theta)$, we need to show that $\sup_{\theta \in \Theta}\left\|\frac{1}{T}\sum_{t=1}^T[g_t(\theta) - m(s_t) \otimes g_*(\theta)]\right\| \xrightarrow{p} 0$. For any $\epsilon > 0$, we have

$$\mathbf{P}\left(\sup_{\theta \in \Theta}\left\|\frac{1}{T}\sum_{t=1}^T[g_t(\theta) - m(s_t) \otimes g_*(\theta)]\right\| < \epsilon\right) \geq \mathbf{P}\left(\sup_{\theta \in \Theta}\sum_{j=1}^M\left|\frac{1}{T}\sum_{t=1}^T[g_{t,j}(\theta) - m_j(s_t)g_*(\theta)_j]\right| < \epsilon\right)$$

$$\overset{(a)}{\geq} 1 - \sum_{j=1}^M \mathbf{P}\left(\sup_{\theta \in \Theta}\left|\frac{1}{T}\sum_{t=1}^T[g_{t,j}(\theta) - m_j(s_t)g_*(\theta)_j]\right| \geq \frac{\epsilon}{M}\right)$$

$$\overset{(b)}{\geq} 1 - o_p(1),$$

$$\therefore\ \sup_{\theta \in \Theta}\left\|\frac{1}{T}\sum_{t=1}^T[g_t(\theta) - m(s_t) \otimes g_*(\theta)]\right\| \xrightarrow{p} 0,$$

where (a) follows by the union bound and (b) by applying Lemma 2 for every $j \in [M]$ (using Condition (ii)). $\qquad\square$

### A.2 Proof of Proposition 2 (Asymptotic normality)

**Proposition 7** (Martingale CLT [15, Corollary 3.1]). *Let $M_i$ with $1 \leq i \leq n$ be a martingale adapted to the filtration $\mathcal{F}_i$ with differences $X_i = M_i - M_{i-1}$ and $M_0 = 0$. Suppose that the following two conditions hold: (i) (Conditional Lindeberg) $\forall\epsilon > 0,\ \sum_{i=1}^n \mathbf{E}\left[X_i^2 I\left(|X_i| > \epsilon\right)|\mathcal{F}_{i-1}\right] \xrightarrow{p} 0$, and (ii) (Convergence of conditional variance) For some constant $\sigma > 0$, $\sum_{i=1}^n \mathbf{E}\left[X_i^2|\mathcal{F}_{i-1}\right] \xrightarrow{p} \sigma^2$. Then $\sum_{i=1}^n X_i \xrightarrow{d} \mathcal{N}(0, \sigma^2)$.*

**Proposition** (Asymptotic normality). *Suppose that (i) $\widehat{\theta}_T^{(\pi)} \xrightarrow{p} \theta^*$; (ii) $\forall(i,j) \in [M] \times [D],\ \left[\frac{\partial \tilde{g}_t}{\partial \theta}(\theta)\right]_{i,j}$ satisfies Property 1; (iii) $\exists \delta > 0$ such that $\mathbf{E}\left[\|\tilde{g}_i(\theta^*)\|^{2+\delta}\right] < \infty$, and (iv) (Selection ratio convergence) $\kappa_T^{(\pi)} \xrightarrow{p} k$ for some constant $k \in \Delta_\psi$. Then $\widehat{\theta}_T$ is asymptotically normal:*

$$\sqrt{T}(\widehat{\theta}_T^{(\pi)} - \theta^*) \xrightarrow{d} \mathcal{N}\left(0, \Sigma(\theta^*, k)\right),$$

where $\Sigma(\theta^*, k)$ is a constant matrix that depends only on $\theta^*$ and $k$. By Assumption 2(e) and the Delta method, $\widehat{\beta}_T$ is asymptotically normal:

$$\sqrt{T}(\widehat{\beta}_T - \beta^*) \xrightarrow{d} \mathcal{N}\left(0, V(\theta^*, k)\right), \text{ where } V(\theta^*, k) = \nabla_\theta f_{tar}(\theta^*)^\top [\Sigma(\theta^*, k)] \nabla_\theta f_{tar}(\theta^*).$$

*Proof.* We follow a standard GMM asymptotic normality proof (e.g. Newey and McFadden [30, Theorem 3.4]) and modify it to work for dependent data. Applying the GMM first-order condition to the two-step GMM estimator, we get

$$\sqrt{T}(\widehat{\theta}_T - \theta^*) = \left[\widehat{G}^\top(\widehat{\theta}_T)\widehat{\Omega}(\widehat{\theta}_T^{(os)})^{-1}\widehat{G}(\tilde{\theta})\right]^{-1} \widehat{G}^\top(\widehat{\theta}_T)\widehat{\Omega}(\widehat{\theta}_T^{(os)})^{-1}\frac{1}{\sqrt{T}}\sum_{t=1}^{T} g_i(\theta^*),$$

where $\widehat{\theta}_T^{(os)}$ is the one-step GMM estimator, $\tilde{\theta}$ is a point on the line-segment joining $\widehat{\theta}_T$ and $\theta^*$,

$$\widehat{G}(\theta) = \frac{1}{T}\sum_{t=1}^{T}\frac{\partial g_t(\theta)}{\partial \theta}$$

$$= \frac{1}{T}\sum_{t=1}^{T}\frac{\partial m(s_t) \otimes \tilde{g}_t(\theta)}{\partial \theta}$$

$$= \frac{1}{T}\sum_{t=1}^{T}\left(\underbrace{[m(s_t), m(s_t), \ldots, m(s_t)]}_{D \text{ times}} \otimes \left[\frac{\partial \tilde{g}_t(\theta)}{\partial \theta}\right]\right),$$

$$= \frac{1}{T}\sum_{t=1}^{T}\left(m_G(s_t) \otimes \left[\frac{\partial \tilde{g}_t(\theta)}{\partial \theta}\right]\right), \text{ and}$$

$$\widehat{\Omega}(\theta) = \frac{1}{T}\sum_{t=1}^{T}\left[g_t(\theta)g_t(\theta)^\top\right]$$

$$= \frac{1}{T}\sum_{t=1}^{T}\left([m(s_t)m(s_t)^\top] \otimes [\tilde{g}_t(\theta)\tilde{g}_t(\theta)^\top]\right),$$

$$= \frac{1}{T}\sum_{t=1}^{T}\left(m_\Omega(s_t) \otimes [\tilde{g}_t(\theta)\tilde{g}_t(\theta)^\top]\right),$$

where $m_G(s_t) = \underbrace{[m(s_t), m(s_t), \ldots, m(s_t)]}_{D \text{ times}}$ is a $M \times D$ matrix and $m_\Omega(s_t) = m(s_t)m(s_t)^\top$.

**Convergence of $\widehat{G}(\widehat{\theta}_T)$.** Let $G(\theta) = \mathbf{E}\left[\frac{\partial \tilde{g}_t(\theta)}{\partial \theta}\right]$. Applying Lemma 2 to every element of $\widehat{G}$ (using Condition (ii)) and using the union bound, we get

$$\sup_{\theta \in \Theta}\left\|\widehat{G}(\theta) - \left(\frac{1}{T}\sum_{t=1}^{T}m_G(s_t)\right) \otimes G(\theta)\right\| \xrightarrow{p} 0,$$

$$\therefore \forall \epsilon > 0, \mathbf{P}\left(\left\|\widehat{G}(\widehat{\theta}_T) - \left(\frac{1}{T}\sum_{t=1}^{T}m_G(s_t)\right) \otimes G(\widehat{\theta}_T)\right\| > \epsilon\right) \to 0. \tag{3}$$

Since $\kappa_T \xrightarrow{p} k$ for some constant $k$ (by Condition (iv)), $\left(\frac{1}{T}\sum_{t=1}^{T}m_G(s_t)\right)$ also converges in probability to a constant matrix. That is, $\frac{1}{T}\sum_{t=1}^{T}m_G(s_t) \xrightarrow{p} m_G^*(k)$ for some constant matrix $m_G^*(k)$ that only depends on $k$. By the continuity of $G$ and the fact that $\widehat{\theta}_T \xrightarrow{p} \theta^*$ (by Condition (i)), we have $G(\widehat{\theta}_T) \xrightarrow{p} G(\theta^*)$. Using these results with Eq. 3, we get

$$\widehat{G}(\widehat{\theta}_T) \xrightarrow{p} m_G^*(k) \otimes G(\theta)$$

$$= G_*(\theta^*, k), \tag{4}$$

$$\text{Similarly, } \widehat{G}(\tilde{\theta}) \xrightarrow[(a)]{p} G_*(\theta^*, k), \tag{5}$$

where $G_*(\theta^*, k) = m_G^*(k) \otimes G(\theta^*)$ and (a) follows because $\tilde{\theta} \overset{P}{\to} \theta^*$.

**Convergence of the weight matrix $\widehat{W}$.**   Let $\Omega(\theta) = \mathbf{E}\left[\tilde{g}_t(\theta)\tilde{g}_t(\theta)^\top\right]$. By applying Lemma 2 to every element of $\widehat{\Omega}$ (using Condition (iii)) and the union bound, we get

$$\sup_{\theta \in \Theta}\left\|\widehat{\Omega}(\theta) - \left(\frac{1}{T}\sum_{t=1}^{T}m_\Omega(s_t)\right) \otimes \Omega(\theta)\right\| \overset{P}{\to} 0,$$

$$\therefore \ \forall \epsilon > 0, \mathbf{P}\left(\left\|\widehat{\Omega}(\widehat{\theta}_T^{(\mathrm{os})}) - \left(\frac{1}{T}\sum_{t=1}^{T}m_\Omega(s_t)\right) \otimes \Omega(\widehat{\theta}_T^{(\mathrm{os})})\right\| > \epsilon\right) \to 0. \tag{6}$$

Since $\kappa_T \overset{p}{\to} k$ for some constant k (by Condition (iv)), $\left(\frac{1}{T}\sum_{t=1}^{T}m_\Omega(s_t)\right) \overset{p}{\to} m_\Omega^*(k)$ for some constant matrix $m_\Omega^*(k)$ that only depends on $k$. By continuity of $\Omega$ and the fact that $\widehat{\theta}_T^{(\mathrm{os})} \overset{p}{\to} \theta^*$ (which follows by Proposition 1), we have $\Omega(\widehat{\theta}_T^{(\mathrm{os})}) \overset{P}{\to} \Omega(\theta^*)$. Using these results with Eq. 6, we get

$$\widehat{\Omega}(\widehat{\theta}_T^{(\mathrm{os})}) \overset{p}{\to} m_\Omega^*(k) \otimes \Omega(\theta^*)$$
$$= \Omega_*(\theta^*, k),$$
$$\therefore \ \widehat{W} = \widehat{\Omega}(\widehat{\theta}_T^{(\mathrm{os})})^{-1} \overset{p}{\to} \Omega_*(\theta^*, k)^{-1}, \tag{7}$$

where $\Omega_*(\theta^*, k) = m_\Omega^*(k) \otimes \Omega(\theta^*)$.

**Asymptotic normality of $\frac{1}{\sqrt{T}}\sum_{t=1}^{T}g_i(\theta^*)$.**   For this part, we use the Cramer-Wold theorem and the martingale CLT in Proposition 7. For any $v \in \mathbf{R}^M$ s.t. $\|v\| = 1$, $\frac{v^\top g_i(\theta^*)}{\sqrt{T}}$ is a MDS because $\mathbf{E}\left[v^\top g_i(\theta^*)|H_{i-1}\right] = v^\top \mathbf{E}\left[g_i(\theta^*)|H_{i-1}\right] = 0$. We now show that the two conditions of Proposition 7 apply to this MDS.

*(i) Conditional Lindeberg:* The Lyapunov condition implies the Lindeberg condition [5, pg. 6]. In our case, the Lyapunov condition is easier to check and we show that it holds. For some $\delta > 0$, we have

$$\frac{1}{T^{1+\delta/2}}\sum_{i=1}^{T}\left|v^\top g_i(\theta^*)\right|^{2+\delta} \overset{(a)}{\le} \frac{1}{T^{1+\delta/2}}\sum_{i=1}^{T}\|v\|^{2+\delta}\|g_i(\theta^*)\|^{2+\delta}$$

$$\overset{(b)}{=} \frac{1}{T^{1+\delta/2}}\sum_{i=1}^{T}\|g_i(\theta^*)\|^{2+\delta}$$

$$\therefore \ \frac{1}{T^{1+\delta/2}}\sum_{i=1}^{T}\mathbf{E}\left[\left|v^\top g_i(\theta^*)\right|^{2+\delta}\big|H_{i-1}\right] \le \frac{1}{T^{1+\delta/2}}\sum_{i=1}^{T}\mathbf{E}\left[\|g_i(\theta^*)\|^{2+\delta}\big|H_{i-1}\right]$$

$$= \frac{1}{T^{1+\delta/2}}\sum_{i=1}^{T}\mathbf{E}\left[\|m(s_i) \otimes \tilde{g}_i(\theta^*)\|^{2+\delta}\right]$$

$$\overset{(c)}{\le} \frac{1}{T^{1+\delta/2}}\sum_{i=1}^{T}\mathbf{E}\left[\|\tilde{g}_i(\theta^*)\|^{2+\delta}\right]$$

$$\overset{(d)}{\to} 0,$$

where (a) follows by Cauchy-Schwarz, (b) because $\|v\| = 1$, (c) because $m(s_i)$ is a binary vector, and (d) because $\mathbf{E}\left[\|\tilde{g}_i(\theta^*)\|^{2+\delta}\right] < \infty$ (by Condition (iii)).

*(ii) Convergence of conditional variance:* The conditional variance can be written as

$$\frac{1}{T}\sum_{t=1}^{T}\mathbf{E}\left[v^{\top}g_t(\theta^*)g_t(\theta^*)^{\top}v\big|H_{i-1}\right] = \frac{1}{T}\sum_{t=1}^{T}v^{\top}\mathbf{E}\left[g_t(\theta^*)g_t(\theta^*)^{\top}\big|H_{i-1}\right]v$$

$$= v^{\top}\left[\left(\frac{1}{T}\sum_{t=1}^{T}m(s_t)m(s_t)^{\top}\right)\otimes\Omega(\theta^*)\right]v$$

$$\xrightarrow[p]{(a)} v^{\top}\left[m^*_{\Omega}(k)\otimes\Omega(\theta^*)\right]v$$

$$= v^{\top}\left[\Omega_*(\theta^*,k)\right]v,$$

where (a) holds because $\kappa_T \xrightarrow{p} k$ (by Condition (iv)). Thus, using Proposition 7, $\forall v \in \mathbf{R}^M$ s.t. $\|v\| = 1$, we have

$$\frac{1}{\sqrt{T}}\sum_{i=1}^{T}v^{\top}g_i(\theta^*)v \xrightarrow{d} \mathcal{N}\left(0, v^{\top}\Omega_*(\theta^*,k)v\right).$$

Thus, by the Cramer-Wold theorem, we get

$$\frac{1}{\sqrt{T}}\sum_{i=1}^{T}g_i(\theta^*) \xrightarrow{d} \mathcal{N}\left(0, \Omega_*(\theta^*,k)\right). \tag{8}$$

**Asymptotic normality of $\widehat{\theta}_T$** By Eqs. 4, 5, 7, and 8, and Slutsky's theorem, we get

$$\sqrt{T}(\widehat{\theta}_T - \theta^*) \xrightarrow{d} \mathcal{N}\left(0, \Sigma(\theta^*,k)\right),$$

$$\text{where } \Sigma(\theta^*,k) = \left[G_*^{\top}(\theta^*,k)\left(\Omega_*(\theta^*,k)^{-1}\right)G_*(\theta^*,k)\right]^{-1}.$$

$\square$

### A.3 Proof of Theorem 1 (Regret of OMS-ETC)

**Lemma 3** (Consistency of $\widehat{k}_t$)**.** *Suppose that Assumption 3 holds. If $\widehat{\theta}_t \xrightarrow{p} \theta^*$, then $\widehat{k}_t \xrightarrow{p} \kappa^*$ where $\widehat{k}_t = \arg\min_{\kappa \in \Delta_\psi} V(\widehat{\theta}_t, \kappa)$.*

*Proof.* By continuity of $V$, compactness of $\Delta_\psi$, and Assumption 3, $\widehat{k}_t \xrightarrow{p} \arg\min_{\kappa \in \Delta_\psi} V(\theta^*, \kappa) = \kappa^*$. $\square$

**Theorem** (Regret of OMS-ETC)**.** *Suppose that (i) Conditions (i)-(iii) of Proposition 2 hold and (ii) Assumption 3 holds. Case (a): For a fixed $e \in (0,1)$, if $\kappa^* \in \tilde{\Delta}$, then the regret converges to zero: $R_\infty(\pi_{ETC}) = 0$. If $\kappa^* \notin \tilde{\Delta}$, then $\pi_{ETC}$ suffers constant regret: $R_\infty(\pi_{ETC}) = r$ for some constant $r > 0$. Case (b): If $e$ depends on $T$ such that $e = o(1)$ and $Te \to \infty$ as $T \to \infty$ (e.g. $e = \frac{1}{\sqrt{T}}$), then $\forall \theta^* \in \Theta$, we have $R_\infty(\pi_{ETC}) = 0$.*

*Proof.* We first analyze Case (a) of the theorem where $e$ is fixed. By Condition (i), $\widehat{\theta}_{Te} \xrightarrow{p} \theta^*$. We have $\widehat{k} = \arg\min_{\kappa \in \Delta_\psi} V(\widehat{\theta}_{Te}, \kappa)$ and therefore $\widehat{k} \xrightarrow{p} \kappa^*$ (by Lemma 3). Thus, if $\kappa^* \in \tilde{\Delta}$, then $\kappa_T \xrightarrow{p} \widehat{k}$ and therefore $\kappa_T \xrightarrow{p} \kappa^*$. Using Proposition 2, we get

$$\sqrt{T}\left(\widehat{\beta}_T - \beta^*\right) \xrightarrow{d} \mathcal{N}\left(0, V(\theta^*, \kappa^*)\right)$$

$$\therefore R_\infty(\pi_{ETC}) = V(\theta^*, \kappa^*) - V(\theta^*, \kappa^*) = 0.$$

If $\kappa^* \notin \tilde{\Delta}$, then $\kappa_T \xrightarrow{p} \bar{\kappa} \neq \kappa^*$, where $\bar{\kappa} = \arg\min_{\kappa \in \tilde{\Delta}} V(\theta^*, \kappa)$. Using Proposition 2, we have

$$\sqrt{T}\left(\widehat{\beta}_T - \beta^*\right) \xrightarrow{d} \mathcal{N}\left(0, V(\theta^*, \bar{\kappa})\right)$$

$$\therefore R_\infty(\pi_{ETC}) = V(\theta^*, \bar{\kappa}) - V(\theta^*, \kappa^*) \overset{(a)}{>} 0,$$

where (a) follows by Condition (ii).

Now we analyze part (b) of the theorem. When $e$ depends on $T$ such that $e = o(1)$, the feasible region converges to the entire simplex: $\tilde{\Delta} \to \Delta_\psi$ as $T \to \infty$. Thus $\kappa_T - \hat{k} \overset{p}{\to} 0$. Furthermore, since $Te \to \infty$ as $T \to \infty$, we have $\hat{k} \overset{p}{\to} \kappa^*$ and therefore $\kappa_T \overset{p}{\to} \kappa^*$. Using Proposition 2, we get the desired result. $\qquad\square$

### A.4 Proof of Lemma 1 (GMM concentration inequality)

**Proposition 8** (MDS concentration inequality [43, Theorem 2.19]). *Let $\{(D_k, \mathcal{F}_k)\}_{k=1}^\infty$ be a MDS, and suppose that $\mathbf{E}\left[\exp\{\lambda D_k\} \,|\mathcal{F}_{k-1}\right] \leq \exp\left\{\frac{\lambda^2 \nu^2}{2}\right\}$ almost surely for any $\lambda < \frac{1}{\alpha}$. Then the sum satisfies the concentration inequality*

$$\mathbf{P}\left(\left|\frac{1}{n}\sum_{k=1}^n D_k\right| > \eta\right) \leq 2\exp\left\{-\frac{n\eta^2}{2\nu^2}\right\} \ \text{if } 0 \leq \eta < \frac{\nu^2}{\alpha}.$$

**Lemma 4** (Uniform law for dependent data). *Let $a_i(\theta) := S_i \tilde{a}(\theta; X_i)$, where $a_i$ is a real-valued function, $S_i \in \{0, 1\}$ is $H_{i-1}$-measurable, and $X_i \overset{iid}{\sim} \mathbf{P}_{\theta^*}$. Let $\tilde{a}_*(\theta) = \mathbf{E}\left[\tilde{a}(\theta; X_i)\right]$. Suppose that $\tilde{a}(\theta)$ satisfies Property 2. Note that $\mathbf{E}\left[a_i(\theta)|H_{i-1}\right] = S_i \tilde{a}_*(\theta)$. Then, for some constant $\delta_0 > 0$ and $\forall \delta \in (0, \delta_0)$,*

$$\mathbf{P}\left(\sup_{\theta \in \Theta}\left|\frac{1}{T}\sum_{i=1}^T \left[a_i(\theta) - S_i \tilde{a}_*(\theta)\right]\right| > \delta\right) < \frac{1}{\delta^D}\exp\left\{-\mathcal{O}\left(T\delta^2\right)\right\}.$$

*Proof.* Let $U = \{\theta_1, \theta_2, \ldots, \theta_N\}$ be a minimal $\delta$-cover of $\Theta$. We have $N \leq \frac{C}{\delta^D}$ for some constant $C$. Let $q : \Theta \to U$ be a function that returns the closest point from the cover: $q(\theta) = \arg\min_{\theta' \in U} \|\theta - \theta'\|$. We have

$$\sup_{\theta \in \Theta}\left|\frac{1}{T}\sum_{i=1}^T \left[a_i(\theta) - S_i \tilde{a}_*(\theta)\right]\right|$$

$$= \sup_{\theta \in \Theta}\left|\frac{1}{T}\sum_{i=1}^T \left[a_i(\theta) - a_i(q(\theta)) + a_i(q(\theta)) - S_i \tilde{a}_*(q(\theta)) + S_i \tilde{a}_*(q(\theta)) - S_i \tilde{a}(\theta)\right]\right|$$

$$\leq \sup_{\theta \in \Theta}\frac{1}{T}\sum_{i=1}^T |a_i(\theta) - a_i(q(\theta))| + \max_{n \in [N]}\left|\frac{1}{T}\sum_{i=1}^T \left[a_i(\theta_n) - S_i \tilde{a}_*(\theta_n)\right]\right| + \sup_{\theta \in \Theta}\frac{1}{T}\sum_{i=1}^T S_i |\tilde{a}_*(q(\theta)) - \tilde{a}_*(\theta)|$$

$$= \sup_{\theta \in \Theta}\frac{1}{T}\sum_{i=1}^T |S_i (\tilde{a}_i(\theta, X_i) - \tilde{a}_i(q(\theta), X_i))| + \max_{n \in [N]}\left|\frac{1}{T}\sum_{i=1}^T \left[a_i(\theta_n) - S_i \tilde{a}_*(\theta_n)\right]\right| + \sup_{\theta \in \Theta}|\tilde{a}_*(q(\theta)) - \tilde{a}_*(\theta)|$$

$$\leq \sup_{\theta \in \Theta}\frac{1}{T}\sum_{i=1}^T |\tilde{a}(\theta, X_i) - \tilde{a}_i(q(\theta), X_i)| + \max_{n \in [N]}\left|\frac{1}{T}\sum_{i=1}^T \left[a_i(\theta_n) - S_i \tilde{a}_*(\theta_n)\right]\right| + \sup_{\theta \in \Theta}|\tilde{a}_*(q(\theta)) - \tilde{a}_*(\theta)|.$$

We now examine the three terms on the RHS one at a time.

**Third term.** By Lipschitzness of $\tilde{a}_*$ (Property 2(i)), we have:

$$\sup_{\theta \in \Theta}|\tilde{a}_*(q(\theta)) - \tilde{a}_*(\theta)| \leq L_1 \sup_{\theta \in \Theta}\|q(\theta) - \theta\| \leq L_1 \delta.$$

**Second term.** We note that it is a sum of a MDS. By Property 2(ii) and Proposition 8, there exists a constant $C_1 > 0$ such that for $\delta \in (0, C_1)$, we have

$$\forall n \in [N], \ \mathbf{P} \left( \left| \frac{1}{T} \sum_{i=1}^{T} [a_i(\theta_n) - S_i \tilde{a}_*(\theta_n)] \right| < \delta \right) > 1 - \exp\left\{-\mathcal{O}\left(T\delta^2\right)\right\}$$

$$\therefore \ \mathbf{P} \left( \max_{n \in [N]} \left| \frac{1}{T} \sum_{i=1}^{T} [a_i(\theta_n) - S_i \tilde{a}_*(\theta_n)] \right| < \delta \right) > 1 - \mathbf{P} \left( \bigcup_{n \in [N]} \left| \frac{1}{T} \sum_{i=1}^{T} [a_i(\theta_n) - S_i \tilde{a}_*(\theta_n)] \right| > \delta \right)$$

$$> 1 - N \exp\left\{-\mathcal{O}\left(T\delta^2\right)\right\}$$

$$> 1 - \frac{1}{\delta^D} \exp\left\{-\mathcal{O}\left(T\delta^2\right)\right\}.$$

**First term.** We have

$$u_*(\eta) = \mathbf{E} \left[ \sup_{\theta, \theta' \in \Theta; \|\theta - \theta'\| \leq \eta} |\tilde{a}_i(\theta, X_i) - \tilde{a}_i(\theta', X_i)| \right]$$

$$\leq \mathbf{E} \left[ \sup_{\theta \in \Theta} \|A(X_i, \theta)\| \sup_{\theta, \theta' \in \Theta; \|\theta - \theta'\| \leq \eta} \|\theta - \theta'\| \right]$$

$$\leq \eta \sup_{\theta \in \Theta} \|A(X_i, \theta)\|$$

$$\overset{(a)}{\leq} A_0 \eta., \tag{9}$$

where (a) follows by Property 2(iii).

Suppose that Property 2(iv)(a) holds. Then

$$\sup_{\theta \in \Theta} \frac{1}{T} \sum_{i=1}^{T} |\tilde{a}_i(\theta, X_i) - \tilde{a}_i(q(\theta), X_i)| \leq \frac{1}{T} \sum_{i=1}^{T} u_i(\delta)$$

$$\leq \frac{1}{T} \sum_{i=1}^{T} u_i(\delta) - u_*(\delta) + u_*(\delta)$$

$$\overset{(a)}{\leq} \frac{1}{T} \sum_{i=1}^{T} u_i(\delta) - u_*(\delta) + A_0 \delta,$$

where (a) follows by Eq. 9. By Property 2(iv)(a), $(u_i(\delta) - u_*(\delta))$ is sub-Exponential. By the sub-exponential tail bound [43, Proposition 2.9], for some constant $C_2 > 0$ and $\delta \in (0, C_2)$, we have

$$\mathbf{P} \left( \left| \frac{1}{T} \sum_{i=1}^{T} u_i(\delta) - u_*(\delta) \right| < \delta \right) > 1 - \exp\left\{-\mathcal{O}\left(T\delta^2\right)\right\}$$

$$\therefore \ \mathbf{P} \left( \sup_{\theta \in \Theta} \frac{1}{T} \sum_{i=1}^{T} |\tilde{a}_i(\theta, X_i) - \tilde{a}_i(q(\theta), X_i)| < (A_0 + 1)\delta \right) > 1 - \exp\left\{-\mathcal{O}\left(T\delta^2\right)\right\}$$

$$\therefore \ \mathbf{P} \left( \sup_{\theta \in \Theta} \frac{1}{T} \sum_{i=1}^{T} |\tilde{a}_i(\theta, X_i) - \tilde{a}_i(q(\theta), X_i)| < \delta \right) > 1 - \exp\left\{-\mathcal{O}\left(T\delta^2\right)\right\}.$$

Now suppose that Property 2(iv)(b) holds instead. Then

$$\sup_{\theta \in \Theta} \frac{1}{T} \sum_{i=1}^{T} |\tilde{a}_i(\theta, X_i) - \tilde{a}_i(q(\theta), X_i)| \leq \frac{1}{T} \sum_{i=1}^{T} \sup_{\theta \in \Theta} \|A(X_i, \theta)\| \sup_{\theta \in \Theta} \|\theta - q(\theta)\|$$

$$\leq \frac{\delta}{T} \sum_{i=1}^{T} \sup_{\theta \in \Theta} \|A(X_i, \theta)\|.$$

Since $\sup_{\theta \in \Theta} \|A(X_i, \theta)\|$ is sub-Exponential, so is $\sum_{i=1}^{T} \sup_{\theta \in \Theta} \|A(X_i, \theta)\|$. By a sub-Exponential tail bound [42, Proposition 2.7.1(a)], we have for any $C_3 > 0$,

$$\mathbf{P}\left(\frac{1}{T}\sum_{i=1}^{T}\sup_{\theta\in\Theta}\|A(X_i,\theta)\| > C_3\right) \leq \exp\left\{-\mathcal{O}\left(TC_3\right)\right\}$$

$$\therefore \ \mathbf{P}\left(\sup_{\theta\in\Theta}\frac{1}{T}\sum_{i=1}^{T}|\tilde{a}_i(\theta,X_i) - \tilde{a}_i(q(\theta),X_i)| > \delta C_3\right) \leq \exp\left\{-\mathcal{O}\left(TC_3\right)\right\}$$

$$\therefore \ \mathbf{P}\left(\sup_{\theta\in\Theta}\frac{1}{T}\sum_{i=1}^{T}|\tilde{a}_i(\theta,X_i) - \tilde{a}_i(q(\theta),X_i)| > \delta\right) \leq \exp\left\{-\mathcal{O}\left(T\right)\right\}.$$

Combining these results together using the union bound, we get

$$\mathbf{P}\left(\sup_{\theta\in\Theta}\left|\frac{1}{T}\sum_{i=1}^{T}[a_i(\theta) - S_i\tilde{a}_*(\theta;k)]\right| < (L_1 + L_2 + 2)\delta\right)$$

$$> \mathbf{P}\left(\max_{n\in[N]}\left|\frac{1}{T}\sum_{i=1}^{T}[a_i(\theta_n) - S_i\tilde{a}_*(\theta_n)]\right| < \delta, \ \left|\frac{1}{T}\sum_{i=1}^{T}u_i(\delta) - u_*(\delta)\right| < \delta\right)$$

$$> 1 - \sum_{n=1}^{N}\mathbf{P}\left(\left|\frac{1}{T}\sum_{i=1}^{T}[a_i(\theta_n) - S_i\tilde{a}_*(\theta_n)]\right| > \delta\right) - \mathbf{P}\left(\left|\frac{1}{T}\sum_{i=1}^{T}u_i(\delta) - u_*(\delta)\right| > \delta\right)$$

$$> 1 - \frac{1}{\delta^D}\exp\left\{-\mathcal{O}\left(T\delta^2\right)\right\}$$

$$\therefore \ \mathbf{P}\left(\sup_{\theta\in\Theta}\left|\frac{1}{T}\sum_{i=1}^{T}[a_i(\theta) - S_i\tilde{a}_*(\theta;k)]\right| < \delta\right) > 1 - \frac{1}{\delta^D}\exp\left\{-\mathcal{O}\left(T\delta^2\right)\right\}.$$

$\square$

**Proposition 9** (Boundedness and Property 2(iv)(a))**.** *Property 2(iv)(a) is satisfied for bounded function classes, i.e., when $\|\tilde{a}_i\|_\infty < A < \infty$.*

*Proof.* We have:

$$u_i(\eta) = \sup_{\theta,\theta'\in\Theta, \|\theta-\theta'\|\leq\eta}|\tilde{a}(\theta,X_i) - \tilde{a}(\theta',X_i)|$$

$$\leq 2\sup_{\theta\in\Theta}|\tilde{a}_i|$$

$$\leq 2A.$$

Thus $u_i(\eta)$ is bounded and therefore sub-Gaussian for every $\eta$. $\square$

**Proposition 10** (Linearity and Property 2(iv)(b))**.** *Suppose that (i) $\tilde{a}(\theta, X_i)$ is a linear function of $\theta$, i.e., $\tilde{a}(\theta, X_i) = \theta^T\phi(X_i) + \rho(X_i)$, where $\phi$ and $\rho$ are arbitrary functions; and (ii) $\forall d \in [D], \ \phi(X_i)_d$ is sub-Exponential. Then $\tilde{a}(\theta, X_i)$ satisfies Property 2(iv)(b).*

*Proof.* We have that $A(X_i, \theta) = \frac{\partial \tilde{a}(X_i; \theta)}{\partial \theta} = \phi(X_i)$ and thus $\sup_{\theta \in \Theta} \|A(X_i, \theta)\| = \|\phi(X_i)\| \leq \sum_{d=1}^{D} |\phi(X_i)_d|$. Therefore, for any $\eta > 0$, we have

$$
\mathbf{P}\left(\sup_{\theta \in \Theta} \|A(X_i, \theta)\| < \eta\right) = \mathbf{P}\left(\|\phi(X_i)\| < \eta\right)
$$

$$
\geq \mathbf{P}\left(\sum_{d=1}^{D} |\phi(X_i)_d| < \eta\right)
$$

$$
\geq \mathbf{P}\left(\forall d \in [D], \ |\phi(X_i)_d| < \frac{\eta}{D}\right)
$$

$$
\overset{(a)}{\geq} 1 - \sum_{d=1}^{D} \mathbf{P}\left(|\phi(X_i)_d| > \frac{\eta}{D}\right)
$$

$$
\overset{(b)}{\geq} 1 - \sum_{d=1}^{D} \exp\left\{-\mathcal{O}(\eta)\right\}
$$

$$
\geq 1 - \exp\left\{-\mathcal{O}(\eta)\right\},
$$

where (a) follows by the union bound and (b) because $\phi(X_i)_d$ is sub-Exponential. This shows that $\sup_{\theta \in \Theta} \|A(X_i, \theta)\|$ is also sub-Exponential (see Vershynin [42, Definition 2.7.5]). □

**Remark.** *Rakhlin et al. [32] derive a uniform martingale LLN and develop sequential analogues of classical complexity measures used in empirical process theory. These techniques are a potential alternative for deriving the tail bound in Lemma 4. However, the conditions required for these techniques are difficult to check. In our case, the dependent and i.i.d. components can be separated more easily. Thus we opted for deriving a uniform concentration bound by modifying the classical uniform LLN proof. Zhan et al. [46] also derive a uniform LLN without requiring boundedness of the martingale difference terms, but with structural assumptions on the summands related to their specific application.*

**Lemma** (GMM concentration inequality)**.** *Let $\lambda_*, C_0, \eta_1, \eta_2,$ and $\delta_0$ be some positive constants. Suppose that (i) Assumption 2 holds; (ii) $\forall j, \tilde{g}_{i,j}(\theta)$ satisfies Property 2; (iii) The spectral norm of the GMM weight matrix $\widehat{W}$ is upper bounded with high probability: $\forall \delta \in (0, C_0), \ \mathbf{P}\left(\|\widehat{W}\| \leq \lambda_*\right) \geq 1 - \frac{1}{\delta^D} \exp\left\{-\mathcal{O}\left(T\delta^2\right)\right\}$ (see Remark 1); (iv) (Local strict convexity) $\forall \theta \in N_{\eta_1}(\theta^*), \ \mathbf{P}\left(\left\|\frac{\partial^2 \bar{Q}}{\partial \theta^2}(\theta)^{-1}\right\| \leq h\right) = 1$ ($\bar{Q}(\theta)$ is defined in Assumption 2(a)); (v) (Strict minimization) $\forall \theta \in N_{\eta_2}(\theta^*),$ there is a unique minimizer $\kappa(\theta) = \arg\min_\kappa V(\theta, \kappa)$ s.t. $V(\theta, \kappa) - V(\theta, \kappa(\theta)) \leq c\delta^2 \implies \|\kappa - \kappa(\theta)\| \leq \delta$; and (vi) $\sup_\kappa |V(\theta, \kappa) - V(\theta', \kappa)| \leq L\|\theta - \theta'\|$. Then, for $\widehat{k}_t = \arg\min_{\kappa \in \Delta_\psi} V(\widehat{\theta}_T^{(\pi)}, \kappa),$ any policy $\pi,$ and $\forall \delta \in (0, \delta_0),$*

$$
\mathbf{P}\left(\left\|\widehat{\theta}_T^{(\pi)} - \theta^*\right\| > \delta\right) < \frac{1}{\delta^{2D}} \exp\left\{-\mathcal{O}\left(T\delta^4\right)\right\} \ \text{and} \ \mathbf{P}\left(\left\|\widehat{k}_T - \kappa^*\right\| > \delta\right) < \frac{1}{\delta^{4D}} \exp\left\{-\mathcal{O}\left(T\delta^8\right)\right\}.
$$

*Proof.* Below we give the empirical and population analogues of the GMM objective for a given policy $\pi$:

Empirical objective: $\widehat{Q}_T^{(\pi)}(\theta) = \left[\frac{1}{T} \sum_{t=1}^{T} g_t(\theta)\right]^\top \widehat{W} \left[\frac{1}{T} \sum_{t=1}^{T} g_t(\theta)\right],$

Population objective: $\bar{Q}_T^{(\pi)}(\theta) = g_T^*(\theta) \widehat{W} g_T^*(\theta)^\top$

$$
= \left[\frac{1}{T} \sum_{t=1}^{T} \mathbf{E}\left[g_t(\theta) | H_{t-1}\right]\right]^\top \widehat{W} \left[\frac{1}{T} \sum_{t=1}^{T} \mathbf{E}\left[g_t(\theta) | H_{t-1}\right]\right]
$$

$$
= \left[\left(\frac{1}{T} \sum_{t=1}^{T} m(s_t)\right) \otimes \tilde{g}_*(\theta)\right]^\top \widehat{W} \left[\left(\frac{1}{T} \sum_{t=1}^{T} m(s_t)\right) \otimes \tilde{g}_*(\theta)\right],
$$

where $\tilde{g}_*(\theta) = \mathbf{E}\left[\tilde{g}_t(\theta)\right].$

To simplify notation, let $m_t = m(s_t)$. By the triangle and Cauchy-Shwartz inequalities (see Newey and McFadden [30, Theorem 2.6]),

$$\left| \widehat{Q}_T^{(\pi)}(\theta) - \bar{Q}_T^{(\pi)}(\theta) \right|$$

$$\leq \left\| \frac{1}{T} \sum_{t=1}^T [g_t(\theta) - m_t \otimes \tilde{g}_*(\theta)] \right\|^2 \|\widehat{W}\|^2 + 2 \|\tilde{g}_*(\theta)\| \left\| \frac{1}{T} \sum_{t=1}^T [g_t(\theta) - m_t \otimes \tilde{g}_*(\theta)] \right\| \|\widehat{W}\|$$

$$\leq \left\| \frac{1}{T} \sum_{t=1}^T [g_t(\theta) - m_t \otimes \tilde{g}_*(\theta)] \right\|^2 \|\widehat{W}\|^2 + 2C \left\| \frac{1}{T} \sum_{t=1}^T [g_t(\theta) - m_t \otimes \tilde{g}_*(\theta)] \right\| \|\widehat{W}\|,$$

where $C = \sup_{\theta \in \Theta} \|\tilde{g}_*(\theta)\|$. By applying Lemma 4 to each element of the vector $g_i(\theta)$ and using the union bound, we get:

$$\mathbf{P}\left( \sup_{\theta \in \Theta} \left\| \frac{1}{T} \sum_{i=1}^T [g_t(\theta) - m_t \otimes \tilde{g}_*(\theta)] \right\| < \delta \right) \geq \mathbf{P}\left( \bigcap_{j=1}^M \sup_{\theta \in \Theta} \left| \frac{1}{T} \sum_{t=1}^T [g_{t,j}(\theta) - m_{t,j} \otimes (\tilde{g}_*)_j(\theta)] \right| < \frac{\delta}{M} \right)$$

$$\geq 1 - \sum_{j=1}^M \mathbf{P}\left( \sup_{\theta \in \Theta} \left| \frac{1}{T} \sum_{t=1}^T [g_{t,j}(\theta) - m_{t,j} \otimes (\tilde{g}_*)_j(\theta)] \right| > \frac{\delta}{M} \right)$$

$$\geq 1 - \frac{1}{\delta^{2D}} \exp\left\{ -\mathcal{O}\left( T\delta^2 \right) \right\}. \tag{10}$$

This means that, for $0 < \delta < 1$,

$$\left\| \frac{1}{T} \sum_{t=1}^T [g_t(\theta) - m_t \otimes \tilde{g}_*(\theta)] \right\| \leq \delta, \|\widehat{W}\| \leq \lambda_* \implies \left| \widehat{Q}_T^{(\pi)}(\theta) - \bar{Q}_T^{(\pi)}(\theta) \right| \leq \lambda_*^2 \delta^2 + 2\lambda_* C\delta$$
$$= (2C + \lambda_* \delta)\lambda_* \delta$$
$$< (2C + \lambda_*)\lambda_* \delta,$$

$$\therefore \ \mathbf{P}\left( \sup_{\theta \in \Theta} \left| \widehat{Q}_T^{(\pi)}(\theta) - \bar{Q}_T^{(\pi)}(\theta) \right| < (2C + \lambda_*)\lambda_* \delta \right) \geq \mathbf{P}\left( \sup_{\theta \in \Theta} \left\| \frac{1}{T} \sum_{t=1}^T [g_t(\theta) - m_t \otimes \tilde{g}_*(\theta)] \right\| \leq \delta, \|\widehat{W}\| \leq \lambda_* \right)$$

$$\overset{(a)}{\geq} 1 - \mathbf{P}\left( \sup_{\theta \in \Theta} \left\| \frac{1}{T} \sum_{t=1}^T [g_t(\theta) - m_t \otimes \tilde{g}_*(\theta)] \right\| > \delta \right) - \mathbf{P}\left( \|\widehat{W}\| > \lambda_* \right)$$

$$\overset{(b)}{\geq} 1 - \frac{1}{\delta^{2D}} \exp\left\{ -\mathcal{O}\left( T\delta^2 \right) \right\}$$

$$\therefore \ \mathbf{P}\left( \sup_{\theta \in \Theta} \left| \widehat{Q}_T^{(\pi)}(\theta) - \bar{Q}_T^{(\pi)}(\theta) \right| < \delta \right) \geq 1 - \frac{1}{\delta^{2D}} \exp\left\{ -\mathcal{O}\left( T\delta^2 \right) \right\},$$

where (a) follows by the union bound and (b) follows by Eq. 10 and Condition (iii). Using this uniform concentration bound, we get

$$\mathbf{P}\left( \bar{Q}_T^{(\pi)}(\widehat{\theta}_T) < \widehat{Q}_T^{(\pi)}(\widehat{\theta}_T) + \frac{\delta}{2} \right) \geq 1 - \frac{1}{\delta^{2D}} \exp\left\{ -\mathcal{O}\left( T\delta^2 \right) \right\},$$

$$\mathbf{P}\left( \widehat{Q}_T^{(\pi)}(\theta^*) < \bar{Q}_T^{(\pi)}(\theta^*) + \frac{\delta}{2} \right) \geq 1 - \frac{1}{\delta^{2D}} \exp\left\{ -\mathcal{O}\left( T\delta^2 \right) \right\}.$$

Since $\widehat{\theta}_T$ minimizes $\widehat{Q}_T^{(\pi)}$ almost surely, we have $\mathbf{P}\left( \widehat{Q}_T^{(\pi)}(\widehat{\theta}_T) \leq \widehat{Q}_T^{(\pi)}(\theta^*) \right) = 1$. Combining these inequalities using the union bound, we get

$$\mathbf{P}\left( \bar{Q}_T^{(\pi)}(\widehat{\theta}_T) < \bar{Q}_T^{(\pi)}(\theta^*) + \delta \right) \geq 1 - \frac{1}{\delta^{2D}} \exp\left\{ -\mathcal{O}\left( T\delta^2 \right) \right\}$$

$$\therefore \ \mathbf{P}\left( \bar{Q}_T^{(\pi)}(\widehat{\theta}_T) < \delta \right) \overset{(a)}{\geq} 1 - \frac{1}{\delta^{2D}} \exp\left\{ -\mathcal{O}\left( T\delta^2 \right) \right\},$$

where (a) follows because $\bar{Q}_T^{(\pi)}(\theta^*) = 0$.

Intuitively, if $\bar{Q}_T^{(\pi)}(\widehat{\theta}_T)$ is small, then we would expect $\widehat{\theta}_T$ to be close to $\theta^*$. To formally show this, we use the local curvature of $\bar{Q}_T^{(\pi)}$. By Condition (iv), $\bar{Q}_T^{(\pi)}$ is locally strictly convex in the $\eta_1$-ball $N_{\eta_1}(\theta^*)$. Therefore, there exists a closed $\gamma$-ball $N_\gamma(\theta^*) \subseteq N_\eta(\theta^*)$ such that

$$\forall \theta \notin N_\gamma(\theta^*), \ \bar{Q}_T^{(\pi)}(\theta) > \bar{Q}_N, \ \text{where} \ \bar{Q}_N = \sup_{\theta \in N_\gamma(\theta^*)} \bar{Q}_T^{(\pi)}(\theta).$$

This is analogous to an identification condition and ensures that $\bar{Q}_T^{(\pi)}(\theta) \leq \bar{Q}_N \implies \theta \in N_\gamma(\theta^*)$.

Let $H(\theta) = \frac{\partial^2 \bar{Q}^{(\pi)}}{\partial \theta^2}(\theta)$. Then, by twice continuous differentiability of $g$, for $\theta \in N_\gamma(\theta^*)$, we have

$$\bar{Q}_T^{(\pi)}(\theta) \overset{(a)}{=} \bar{Q}_T^{(\pi)}(\theta^*) + (\theta - \theta^*) \left[ H(\theta') \right] (\theta - \theta^*)^\top$$

$$\overset{(b)}{=} (\theta - \theta^*) \left[ H(\theta') \right] (\theta - \theta^*)^\top,$$

$$\therefore \ \| \theta - \theta^* \|^2 \leq \bar{Q}_T^{(\pi)}(\theta) \| H^{-1}(\theta') \|$$

$$\overset{(c)}{\leq} \left[ \bar{Q}_T^{(\pi)}(\theta) \right] h,$$

where in (a), $\theta'$ is a point on the line segment joining $\theta$; (b) follows because $\bar{Q}_T^{(\pi)}(\theta^*) = 0$; and (c) follows by Condition (iv). Thus, for $\delta < \bar{Q}_N$, we have

$$\bar{Q}_T^{(\pi)}(\widehat{\theta}_T) < \delta \implies \| \widehat{\theta}_T - \theta^* \| < \sqrt{\delta h}$$

$$\therefore \ \mathbf{P}\left( \| \widehat{\theta}_T - \theta^* \| < \delta \right) \geq \mathbf{P}\left( \bar{Q}_T^{(\pi)}(\widehat{\theta}_T) < \frac{\delta^2}{h} \right)$$

$$\geq 1 - \frac{1}{\delta^{2D}} \exp\left\{ -\mathcal{O}\left( T\delta^4 \right) \right\}.$$

**Concentration inequality for $\widehat{k}_T$**

By Condition (vi), $\sup_{\kappa \in \Delta_\psi} |V(\widehat{\theta}_T, \kappa) - V(\theta^*, \kappa)| \leq L \| \widehat{\theta}_T - \theta^* \|$. Therefore,

$$\| \widehat{\theta}_T - \theta^* \| < \delta \implies \sup_{\kappa \in \Delta_\psi} |V(\widehat{\theta}_T, \kappa) - V(\theta^*, \kappa)| \leq L\delta.$$

Furthermore, we have

$$\sup_{\kappa \in \Delta_\psi} |V(\widehat{\theta}_T, \kappa) - V(\theta^*, \kappa)| \leq L\delta \implies V(\theta^*, \widehat{k}_T) < V(\widehat{\theta}_T, \widehat{k}_T) + L\delta, \ \text{and}$$

$$V(\widehat{\theta}_T, \kappa^*) < V(\theta^*, \kappa^*) + L\delta.$$

Since $\widehat{k}_T$ is the minimizer, we have $V(\widehat{\theta}_T, \widehat{k}_T) \leq V(\widehat{\theta}_T, \kappa^*)$. Combining these inequalities, we get

$$\| \widehat{\theta}_T - \theta^* \| < \delta \implies V(\theta^*, \widehat{k}_T) - V(\theta^*, \kappa^*) < 2L\delta.$$

Due to Condition (v), we have

$$V(\theta^*, \widehat{k}_T) - V(\theta^*, \kappa^*) < 2L\delta \implies \| \widehat{k}_T - \kappa^* \| < \sqrt{\frac{2L\delta}{c}},$$

$$\therefore \ \| \widehat{\theta}_T - \theta^* \| < \delta \implies \| \widehat{k}_T - \kappa^* \| < \sqrt{\frac{2L\delta}{c}}$$

$$\therefore \ \mathbf{P}(\| \widehat{k}_T - \kappa^* \| < \delta) > 1 - \mathbf{P}\left( \| \widehat{\theta}_T - \theta^* \| < \mathcal{O}\left( \delta^2 \right) \right)$$

$$> 1 - \frac{1}{\delta^{4D}} \exp\left\{ -\mathcal{O}\left( T\delta^8 \right) \right\}.$$

$\square$

**Lemma 5** (Sufficient condition for $\widehat{W}$). *Suppose that $\forall (j, k)$, $[\tilde{g}_{i,j}(\theta)\tilde{g}_{i,k}(\theta)]$ satisfies Property 2. Let $\widehat{W}_T(\widehat{\theta}_T^{(os)}) = \widehat{\Omega}_T(\widehat{\theta}_T^{(os)})^{-1} = \left[ \frac{1}{T} \sum_{t=1}^T g_t(\widehat{\theta}_T^{(os)}) g_t^\top(\widehat{\theta}_T^{(os)}) \right]^{-1}$, where $\widehat{\theta}_T^{(os)}$ is the one-step GMM estimator (that uses $\widehat{W} = I$). Then $\widehat{W}_T(\widehat{\theta}_T^{(os)})$ satisfies Condition (iii) of Lemma 1.*

*Proof.* We define $W_T(\theta^*)$ as

$$W_T(\theta^*) = \Omega_T(\theta^*)^{-1} = \left[ \frac{1}{T} \sum_{t=1}^{T} \mathbf{E}\left[ g_t(\theta^*) g_t^\top(\theta^*) \mid H_{t-1} \right] \right]^{-1}$$

$$= \left[ \left( \frac{1}{T} \sum_{t=1}^{T} m(s_t) m^\top(s_t) \right) \otimes \mathbf{E}\left[ \tilde{g}_t(\theta^*) \tilde{g}_t^\top(\theta^*) \right] \right]^{-1}.$$

Let $\Delta = \widehat{\Omega}_T(\widehat{\theta}_T^{(\text{os})}) - \Omega_T(\widehat{\theta}_T^{(\text{os})})$ and $\lambda_{\min}$ denote smallest eigenvalue. Using the eigenvalue stability inequality [38, Section 1.3.3], we get:

$$\left| \lambda_{\min}\left( \widehat{\Omega}_T(\widehat{\theta}_T^{(\text{os})}) \right) - \lambda_{\min}\left( \Omega_T(\widehat{\theta}_T^{(\text{os})}) \right) \right| \le \|\Delta\|,$$

$$\therefore \left\| \widehat{W}_T(\widehat{\theta}_T^{(\text{os})}) \right\| = \left\| \widehat{\Omega}_T(\widehat{\theta}_T^{(\text{os})})^{-1} \right\| = \frac{1}{\lambda_{\min}\left( \widehat{\Omega}_T(\widehat{\theta}_T^{(\text{os})}) \right)} \le \frac{1}{\lambda_{\min}\left( \Omega_T(\widehat{\theta}_T^{(\text{os})}) \right) - \|\Delta\|}. \quad (11)$$

By applying Lemma 4 to each term of the matrix and using the union bound, we have

$$\mathbf{P}\left( \sup_{\theta \in \Theta} \left\| \widehat{\Omega}_T(\theta) - \Omega_T(\theta) \right\| \le \delta \right) \overset{(a)}{\ge} \mathbf{P}\left( \sup_{\theta \in \Theta} \left\| \widehat{\Omega}_T(\theta) - \Omega_T(\theta) \right\|_F \le \delta \right)$$

$$\ge \mathbf{P}\left( \sup_{\theta \in \Theta} \sum_{i,j} \left| \widehat{\Omega}_{T,i,j}(\theta) - \Omega_{T,i,j}(\theta) \right| \le \delta \right)$$

$$\ge 1 - \sum_{i,j} \mathbf{P}\left( \sup_{\theta \in \Theta} \left| \widehat{\Omega}_{T,i,j}(\theta) - \Omega_{T,i,j}(\theta) \right| > \frac{\delta}{M^2} \right)$$

$$= 1 - \frac{1}{\delta^D} \exp\left\{ -\mathcal{O}\left( T\delta^2 \right) \right\}$$

$$\therefore \mathbf{P}\left( \|\Delta\| \le \delta \right) = \mathbf{P}\left( \left\| \widehat{\Omega}_T(\tilde{\theta}_T) - \Omega(\tilde{\theta}_T) \right\| \le \delta \right) \ge 1 - \frac{1}{\delta^D} \exp\left\{ -\mathcal{O}\left( T\delta^2 \right) \right\}, \quad (12)$$

where in (a) $\|.\|_F$ denotes the Frobenius norm.

For some $\delta_0 > 0$, let $\bar{\lambda} = \inf_{\theta \in N_{\delta_0}(\theta^*), \kappa \in \Delta_\psi} \lambda_{\min}\left( \Omega_T(\theta) \right)$. For $\delta \le \min\left\{ \delta_0, \frac{\bar{\lambda}}{2} \right\}$, we have

$$\|\Delta\| \le \delta \overset{(a)}{\implies} \left\| \widehat{W}_T(\tilde{\theta}_T) \right\| \le \frac{2}{\bar{\lambda}},$$

$$\therefore \mathbf{P}\left( \left\| \widehat{W}_T(\tilde{\theta}_T) \right\| \le \frac{2}{\bar{\lambda}} \right) \ge \mathbf{P}\left( \|\Delta\| \le \delta \right)$$

$$\overset{(b)}{\ge} 1 - \frac{1}{\delta^D} \exp\left\{ -\mathcal{O}\left( T\delta^2 \right) \right\},$$

where (a) follows by Eq. 11 and (b) by Eq. 12. $\qquad \square$

In the next lemma, we present a concentration inequality for $\widehat{k}_T$ with better rates under additional restrictions on $\theta^*$. We do not require these better rates for proving zero regret for OMS-ETG. We present this lemma for the sake of completeness.

**Lemma 6** (Another concentration inequality for $\widehat{k}_T$). *Let* $\kappa(\theta) = \arg\min_\kappa V(\theta, \kappa)$, $\Theta_{boundary} = \{\theta \in \Theta : \kappa(\theta) \in boundary(\Delta_\psi)\}$, *where* $boundary(\Delta_\psi) = \{\kappa \in \Delta_\psi : \exists i, \ s.t. \ \kappa_i = 0\}$, $\Theta_{minima} = \{\theta \in \Theta : \frac{\partial V}{\partial \kappa}(\theta, \kappa(\theta)) = 0\}$, *and* $\Theta_{restricted} = \Theta \setminus (\Theta_{boundary} \bigcap \Theta_{minima})$ *Suppose that (i) the conditions of Lemma 1 hold, and (ii)* $\theta \in \Theta_{restricted}$. *Then*

$$\mathbf{P}\left( \left\| \widehat{k}_T - \kappa^* \right\| > \delta \right) < \frac{1}{\delta^{2D}} \exp\left\{ -\mathcal{O}\left( T\delta^4 \right) \right\}.$$

*This means that if* $\theta^*$ *is not such that the minimizer* $\kappa(\theta) = \arg\min_\kappa V(\theta, \kappa)$ *is on the boundary of the simplex and is also a local minimum of* $V(\theta, \kappa)$ *(informally,* $\kappa(\theta)$ *is not "just" on the boundary), we can get better rates.*

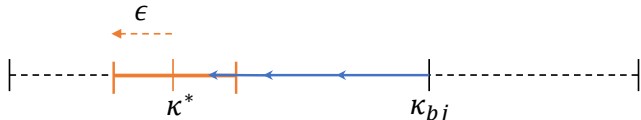

Figure 5: Illustration of the proof of OMS-ETG algorithm. When the event $\mathcal{I}(\epsilon)$ occurs, (a) if the selection ratio $\kappa_{bj}$ is outside $N_\epsilon(\kappa^*)$, then then selection ratio in the next round $\kappa_{b(j+1)}$ will move closer to $N_\epsilon(\kappa^*)$, and (b) if $\kappa_{bj}$ is inside $N_\epsilon(\kappa^*)$, it remains inside for all future rounds.

*Proof.* Now we use the tail bound for $\widehat{\theta}_T$ to derive a concentration inequality for $\widehat{k}_T$ when $\theta \in \Theta_{\text{restricted}}$. $\widehat{k}_T$ is the solution to the following constrained optimization problem:

$$\min_{\kappa \in \mathbf{R}^{|\psi|}} V\left(\widehat{\theta}_T, \kappa\right) \text{ subject to } \sum_{i=1}^{|\psi|} \kappa_i = 1.$$

The Lagrangian function is

$$\mathcal{L}\left(\theta, \kappa, \lambda\right) = V\left(\theta, \kappa\right) + \lambda\left(\sum_{i=1}^{|\psi|} \kappa_i - 1\right).$$

Let $f(\theta, \kappa, \lambda) = \frac{\partial \mathcal{L}}{\partial \kappa}(\theta, \kappa, \lambda) = \frac{\partial V}{\partial \kappa}(\theta, \kappa) + \lambda[1, 1, \ldots, 1]^\top$. Since $\lambda[1, 1, \ldots, 1]^\top \neq 0$, there exists a Lagrange multiplier $\lambda^* \in \mathbf{R}$ such that $f(\theta^*, \kappa^*, \lambda^*) = 0$.

Condition (ii) is required to ensure that $f(\theta, \kappa, \lambda^*)$ is continuously differentiable in $(\theta, \kappa)$ which allows us to use the implicit function theorem. To show this, we divide the space $\Theta_{\text{restricted}}$ into two disjoint sets: (i) $\Theta_{\text{interior}} = \Theta \setminus \Theta_{\text{boundary}}$, and (ii) $\Theta_{\text{strict-boundary}} = \Theta_{\text{boundary}} \bigcap \Theta_{\text{minima}}^c$. When $\theta \in \Theta_{\text{interior}}$, the constraint will *not* be active and thus $\lambda^* = 0$. When $\theta \in \Theta_{\text{strict-boundary}}$, the constraint will be active and thus $\lambda^* > 0$. In both cases, $f(\theta, \kappa, \lambda^*)$ will be continuously differentiable in $(\theta, \kappa)$. Note that if $\theta \in \Theta \setminus \Theta_{\text{restricted}}$, then $\lambda^* = 0$ but $f$ is not differentiable because the constraint is "just" inactive.

Let $Y(\theta, \kappa) = \frac{\partial f}{\partial \kappa}(\theta, \kappa) = \frac{\partial^2 V}{\partial \kappa^2}(\theta, \kappa)$, and $X(\theta, \kappa) = \frac{\partial f}{\partial \theta}(\theta, \kappa) = \frac{\partial^2 V}{\partial \theta \partial \kappa}(\theta, \kappa)$. By the implicit function theorem, since $Y(\theta^*, \kappa^*)$ is invertible (by Condition (v)), there exist neighbourhoods $N(\theta^*)$ and $N(\kappa^*)$ and a function $\phi : N(\theta^*) \to N(\kappa^*)$ such that $\widehat{k}_T = \phi(\widehat{\theta}_T)$ and $\frac{\partial \phi}{\partial \theta}(\theta) = -\left[Y(\theta, \phi(\theta))^{-1} X(\theta, \phi(\theta))\right]$. By a Taylor expansion, we get

$$\widehat{k}_T = \phi(\widehat{\theta}_T) \overset{(a)}{=} \phi(\theta^*) + \frac{\partial \phi}{\partial \theta}(\tilde{\theta})\left(\widehat{\theta}_T - \theta^*\right)$$

$$= \kappa^* + \frac{\partial \phi}{\partial \theta}(\tilde{\theta})\left(\widehat{\theta}_T - \theta^*\right)$$

$$\therefore \left\|\widehat{k}_T - \kappa^*\right\| \leq \left\|\frac{\partial \phi}{\partial \theta}(\tilde{\theta})\right\| \left\|\widehat{\theta}_T - \theta^*\right\|$$

$$\leq C \left\|\widehat{\theta}_T - \theta^*\right\|,$$

where in (a) $\tilde{\theta}$ is a point on the line segment joining $\widehat{\theta}_T$ and $\theta^*$, and $C = \sup_{\theta \in \mathcal{N}(\theta^*)} \left\|\frac{\partial \phi}{\partial \theta}(\theta)\right\|$. Therefore, we have

$$\mathbf{P}\left(\|\widehat{\kappa}_T - \kappa^*\| \leq \delta\right) \geq \mathbf{P}\left(\left\|\widehat{\theta}_T - \theta^*\right\| \leq \frac{\delta}{C}\right) \geq 1 - \frac{1}{\delta^{2D}} \exp\left\{-\mathcal{O}\left(t\delta^4\right)\right\}.$$

$\square$

## A.5   Proof of Theorem 2 (Regret of OMS-ETG)

**Theorem** (Regret of OMS-ETG). *Suppose that Conditions (i)-(iv) of Proposition 2 hold. Let* $\tilde{\Delta}(s) = \{s\kappa_b + (1-s)\kappa : \kappa \in \Delta_\psi\}$. *Case (a): For a fixed $s \in (0, 1)$, if the oracle selection ratio*

$\kappa^* \in \tilde{\Delta}(s)$, then the regret converges to zero: $R_\infty(\pi_{ETG}) = 0$. If $\kappa^* \notin \tilde{\Delta}(s)$, then $R_\infty(\pi_{ETG}) = r$ *for some constant $r > 0$. Case (b): Now also suppose that the conditions for Lemma 1 hold. If $s = CT^{\eta-1}$ for some constant $C$ and any $\eta \in [0, 1)$, then $\forall \theta^* \in \Theta$, the regret converges to zero: $R_\infty(\pi_{ETG}) = 0$.*

*Proof.* We prove this theorem by first showing that $\kappa_T \xrightarrow{p} \kappa^*$. Then we can apply Proposition 2 to get the desired result. Recall that $b = Ts$ is the batch size.

**Case 1: when $s \in (0, 1)$ is a fixed constant and $\kappa^* \in \tilde{\Delta}(s)$.**

Let $\mathcal{I}(\epsilon)$ be the event that $\widehat{k}_{bj}$ remains inside an $\epsilon$-ball of $\kappa^*$ (denoted by $N_\epsilon(\kappa^*)$) for all rounds $j \in [J]$. That is, $\mathcal{I}(\epsilon) = \left\{ \forall j \in [J], \widehat{k}_{bj} \in N_\epsilon(\kappa^*) \right\}$. If $\kappa^* \in \tilde{\Delta}(s)$, then to prove that $\kappa_T \xrightarrow{p} \kappa^*$, it is sufficient to show that $\forall \epsilon > 0$, $\mathcal{I}(\epsilon)$ occurs w.p.a. 1.

This is because in OMS-ETG, after every round, we move as close to $\widehat{k}_{bj}$ as possible. This is illustrated in Figure 5 for the case when $\Delta_\psi$ is a 1-simplex. When $\mathcal{I}(\epsilon)$ occurs, if the selection ratio $\kappa_{bj}$ after round $j$ is outside $N_\epsilon(\kappa^*)$, we move towards it in the subsequent round and thus $\kappa_{b(j+1)}$ will be closer to $N_\epsilon(\kappa^*)$. Once the selection ratio enters $N_\epsilon(\kappa^*)$ (which it is guaranteed to if $\kappa^* \in \tilde{\Delta}(s)$), it will remain inside $N_\epsilon(\kappa^*)$ for every round after that. Thus $\mathcal{I}(\epsilon) \implies \kappa_T \in N_\epsilon(\kappa^*)$. Therefore, we have

$$\forall \epsilon > 0, \ \mathbf{P}(\kappa_T \in N_\epsilon(\kappa^*)) \geq \mathbf{P}(\mathcal{I}(\epsilon))$$
$$= \mathbf{P}\left( \forall j \in [J], \ \widehat{k}_{bj} \in N_\epsilon(\kappa^*) \right)$$
$$= 1 - \mathbf{P}\left( \bigcup_{j=1}^{J} \left\| \widehat{k}_{bj} - \kappa^* \right\| > \epsilon \right)$$
$$\overset{(a)}{\geq} 1 - \sum_{j=1}^{J} \mathbf{P}\left( \left\| \widehat{k}_{bj} - \kappa^* \right\| > \epsilon \right)$$
$$\overset{(b)}{\longrightarrow} 1,$$
$$\therefore \ \kappa_T \xrightarrow{p} \kappa^*,$$

where (a) follows by the union bound and (b) follows because $J$ is finite and $\forall j, \ \widehat{k}_{bj} \xrightarrow{p} \kappa^*$ (by Lemma 3).

**Case 2: when $s$ depends on the horizon $T$.**

**Case 2(a): when $s \in \Omega(T^{\eta-1})$ for any $\eta \in (0, 1)$.**

Similar to Case 1, it is sufficient to show that the event $\mathcal{I}(\epsilon) = \left\{ \forall j \in [J], \ \widehat{k}_{bj} \in N_\epsilon(\kappa^*) \right\}$ occurs w.p.a. 1 for every $\epsilon > 0$. However, since $J \to \infty$, consistency of $\widehat{k}_{bj}$ is no longer sufficient to prove

this. Instead, we use the concentration inequality in Lemma 1:

$$\forall \epsilon > 0, \ \mathbf{P}(\kappa_T \in N_\epsilon(\kappa^*)) \geq \mathbf{P}(\mathcal{I}(\epsilon))$$

$$= \mathbf{P}\left(\forall j \in [J], \ \widehat{k}_{bj} \in N_\epsilon(\kappa^*)\right)$$

$$= \mathbf{P}\left(\forall j \in [J], \ \left\|\widehat{k}_{bj} - \kappa^*\right\| \leq \epsilon\right)$$

$$= 1 - \mathbf{P}\left(\bigcup_{j=1}^{J} \left\|\widehat{k}_{bj} - \kappa^*\right\| > \epsilon\right)$$

$$\overset{(a)}{\geq} 1 - \sum_{j=1}^{J} \mathbf{P}\left(\left\|\widehat{k}_{bj} - \kappa^*\right\| > \epsilon\right)$$

$$\overset{(b)}{\geq} 1 - \sum_{j=1}^{J} \frac{1}{\epsilon^{4D}} \exp\left\{-\mathcal{O}\left(-Tsj\epsilon^8\right)\right\}$$

$$\overset{(c)}{\geq} 1 - \sum_{j=1}^{J} \frac{1}{\epsilon^{4D}} \exp\left\{-\mathcal{O}\left(-Ts\epsilon^8\right)\right\}$$

$$= 1 - \frac{J}{\epsilon^{4D}} \exp\left\{-\mathcal{O}\left(-Ts\epsilon^8\right)\right\}$$

$$= 1 - \frac{1}{s\epsilon^{4D}} \exp\left\{-\mathcal{O}\left(-Ts\epsilon^8\right)\right\}$$

$$\to 1 \text{ if } s = CT^{\eta-1}$$

for any $\eta \in (0, 1)$ and some constant $C$. Here (a) follows by the union bound, (b) by Lemma 1, and (c) because $j \geq 1$.

**Case 2(b): when $s = \frac{C}{T}$ for some constant $C > 0$.**

We prove this similarly to Case 2(a). However, in this case, the number of rounds $J = \frac{1}{s} \in \mathcal{O}(T)$. Let $f = T^{\gamma-1}$ for some $\gamma \in (0, 1)$ and $\mathcal{I}(f, \epsilon) = \left\{\forall j \in [Jf+1, \ldots, J], \ \widehat{k}_{bj} \in N_\epsilon(\kappa^*)\right\}$ be the event that $\widehat{k}_{bj}$ remains inside $N_\epsilon(\kappa^*)$ *after* the first $Jf$ rounds.

Since $f \in o(1)$, we have $\mathcal{I}(f, \epsilon) \implies \kappa_T \in N_\epsilon(\kappa^*)$ for every $\epsilon > 0$. This is because the fraction $f$ is asymptotically negligible and thus we can effectively ignore the first $Jf$ rounds. Therefore we have

$$\forall \epsilon > 0, \ \mathbf{P}(\kappa_T \in N_\epsilon(\kappa^*)) \geq \mathbf{P}(\mathcal{I}(f, \epsilon))$$

$$= \mathbf{P}\left(\forall j \in [Jf + 1, Jf + 2, \ldots, J], \left\|\widehat{k}_{bj} - \kappa^*\right\| \leq \epsilon\right)$$

$$= 1 - \mathbf{P}\left(\bigcup_{j=Jf+1}^{J} \left\|\widehat{k}_{bj} - \kappa^*\right\| > \epsilon\right)$$

$$\geq 1 - \sum_{j=Jf}^{J} \mathbf{P}\left(\left\|\widehat{k}_{bj} - \kappa^*\right\| > \epsilon\right)$$

$$\geq 1 - \sum_{j=Jf+1}^{J} \frac{1}{\epsilon^{4D}} \exp\left\{-\mathcal{O}\left(Tsj\epsilon^8\right)\right\}$$

$$\overset{(a)}{\geq} 1 - \sum_{j=Jf+1}^{J} \frac{1}{\epsilon^{4D}} \exp\left\{-\mathcal{O}\left(j\epsilon^8\right)\right\}$$

$$\overset{(b)}{\geq} 1 - \sum_{j=Jf+1}^{J} \frac{1}{\epsilon^{4D}} \exp\left\{-\mathcal{O}\left(Jf\epsilon^8\right)\right\}$$

$$\overset{(c)}{\geq} 1 - \sum_{j=Jf+1}^{J} \frac{1}{\epsilon^{4D}} \exp\left\{-\mathcal{O}\left(T^\gamma \epsilon^8\right)\right\}$$

$$\geq 1 - \frac{J}{\epsilon^{4D}} \exp\left\{-\mathcal{O}\left(T^\gamma \epsilon^8\right)\right\}$$

$$\geq 1 - \frac{T}{\epsilon^{4D}} \exp\left\{-\mathcal{O}\left(T^\gamma \epsilon^8\right)\right\}$$

$$\to 1,$$

where (a) follows because $Ts = C$, (b) because $j \geq Jf$, and (c) because $Jf = \mathcal{O}(T^\gamma)$. We note that it is possible to unify the analysis of Case 2(a) and Case 2(b) by ignoring the first $Jf$ rounds in Case 2(a) as well. We prove the two cases separately for the sake of clarity.

$\square$

## B   Incorporate Cost Structure

### B.1   Proof of Proposition 3 (Regret of OMS-ETC-CS)

**Proposition** (Regret of OMS-ETC-CS). *Suppose that the conditions of Theorem 1 hold. If $e = o(1)$ such that $Be \to \infty$ as $B \to \infty$, then $\forall \theta^* \in \Theta$, $R_\infty(\pi_{ETC\text{-}CS}) = 0$.*

*Proof.* The proof is almost exactly like that of Theorem 1. We prove that $\kappa_T \overset{p}{\to} \kappa^*$ and then apply Proposition 2. Let the number of samples used for exploration be $T_e$. Since $\kappa_{T_e} = \left[\frac{1}{|\psi|}, \frac{1}{|\psi|}, \ldots, \frac{1}{|\psi|}\right]$, we have

$$T_e = \frac{Be}{\kappa_{T_e}^\top c}.$$

$T_e$ is not a random variable because $\kappa_{T_e}$ is fixed. By Lemma 3, we have $\widehat{k}_{T_e} \overset{p}{\to} \kappa^*$.

When $e \in o(1)$, the feasible region converges to the entire simplex, i.e., $\tilde{\Delta} \to \Delta_\psi$. Thus $\kappa_T - \widehat{k}_{T_e} \overset{p}{\to} 0$. $\square$

**Input:** $B, s, c$

1 $\widehat{k} = \text{ctr}\left(\Delta_\psi\right)$ ;
2 $b = \frac{Bs}{c_{\max}}$ ;
3 $B_l = B$;
4 $j = 0$ ;
5 **while** $B_l > 0$ **do**
6 $\quad j \leftarrow j + 1$;
7 $\quad$ last_step $= \frac{B_l}{c_{\max}} \leq b$;
8 $\quad$ **if** *not last_step* **then**
9 $\quad\quad$ Collect $b$ samples s.t. $\kappa_{bj} = \widehat{k}$;
10 $\quad\quad B_l \leftarrow B - bj\left(\kappa_{bj}^\top c\right)$;
11 $\quad\quad t = b(j+1)$;
12 $\quad\quad \widehat{\theta}_t = \text{GMM}(H_t, \widehat{W} = \widehat{W}_{\text{valid}})$;
13 $\quad\quad \widehat{k}_t = \arg\min_{\kappa \in \Delta_\psi} V(\widehat{\theta}_t, \kappa)\left(\kappa^\top c\right)$;
14 $\quad\quad \widehat{k} = \text{proj}(\widehat{k}_{\min}, \tilde{\Delta}_{j+1}(\kappa_t))$;
15 $\quad$ **else**
16 $\quad\quad$ Collect samples s.t. $\kappa_T = \widehat{k}$;
17 $\quad\quad B_l \leftarrow 0$;
18 $\quad$ **end**
19 **end**
20 $\widehat{\theta}_T = \text{GMM}(H_T, \widehat{W} = \widehat{W}_{\text{efficient}})$;

**Output:** $\widehat{\theta}_T$

(a) OMS-ETG-FS (fixed samples per batch).

**Input:** $B, s, c$

1 $\widehat{k} = \text{ctr}\left(\Delta_\psi\right)$;
2 $J = \frac{B}{s}$;
3 $t = 0$;
4 **for** $j \in [1, 2, \ldots, J]$ **do**
5 $\quad b = \frac{Bs}{(\widehat{k}^\top c)}$;
6 $\quad t \leftarrow t + b$;
7 $\quad$ Collect $b$ samples s.t. $\kappa_t = \widehat{k}$;
8 $\quad \widehat{\theta}_t = \text{GMM}(H_t, \widehat{W} = \widehat{W}_{\text{valid}})$;
9 $\quad \widehat{k}_t = \arg\min_{\kappa \in \Delta_\psi} V(\widehat{\theta}_t, \kappa)\left(\kappa^\top c\right)$;
10 $\quad \widehat{k} = \text{proj}(\widehat{k}_{\min}, \tilde{\Delta}_{j+1}(\kappa_t))$;
11 **end**
12 $\widehat{\theta}_T = \text{GMM}(H_T, \widehat{W} = \widehat{W}_{\text{efficient}})$;

**Output:** $\widehat{\theta}_T$

(b) OMS-ETG-FB (fixed budget per batch)

Figure 6: Algorithms for OMS-ETG-FS and OMS-ETG-FB.

## B.2 Proof of Proposition 4 (Regret of OMS-ETG-FS)

**Proposition** (Regret of OMS-ETG-FS). *Suppose that the conditions of Theorem 2 hold. If $s = B^{\eta-1}$ and any $\eta \in [0, 1)$, then $\forall \theta^* \in \Theta$, $R_\infty\left(\pi_{ETG\text{-}FS}\right) = 0$.*

*Proof.* We can prove this similarly to Theorem 2. The key difference is that the number of rounds $J$ is now a random variable. But we can use the fact the $J$ is bounded:

$$\frac{1}{s} \leq J \leq \frac{c_{\max}}{sc_{\min}},$$
$$\therefore \ J \in \mathcal{O}\left(\frac{1}{s}\right).$$

Now we can proceed like Case 2 in the proof of Theorem 2. $\qquad\square$

## B.3 Proof of Proposition 5 (Regret of OMS-ETG-FB)

**Proposition** (Regret of OMS-ETG-FB). *Suppose that the conditions of Theorem 2 hold. If $s = B^{\eta-1}$ and any $\eta \in [0, 1)$, then $\forall \theta^* \in \Theta$, $R_\infty\left(\pi_{ETG\text{-}FB}\right) = 0$.*

*Proof.* We show this similarly to Theorem 2. In this case, the size of each batch is random but the numbers of rounds $J = \frac{1}{s}$ is not random. Thus we can't use the concentration inequality in Lemma 1 directly since that only holds for a fixed time step $t$. We get around this by showing that the estimated selection ratio $\widehat{k}_t$ will remain in an $\epsilon$-ball around $\kappa^*$ uniformly over all time steps after some asymptotically negligible fraction of the horizon $T$.

Let $T_j$ be the number of samples collected after round $j$, i.e., $T_j = \frac{Bsj}{\kappa_{T_j}^\top c}$. Let $f = B^{\gamma-1}$ for some $\gamma \in (0, 1)$. Like the proof of Theorem 2, we can ignore the first $Jf$ rounds since they are $f \in o(1)$ is

an asymptotically negligible fraction. And similarly to the proof of Theorem 2, in order to show that $\kappa_T \xrightarrow{p} \kappa^*$, it is sufficient to show that $\mathbf{P}\left(\forall j \in [Jf+1, Jf+2, \ldots, J], \left\|\widehat{k}_{T_j} - \kappa^*\right\| \leq \epsilon\right) \xrightarrow{B \to \infty} 1$. We can show this as follows:

$$\mathbf{P}\left(\forall j \in [Jf+1, Jf+2, \ldots, J], \left\|\widehat{k}_{T_j} - \kappa^*\right\| \leq \epsilon\right) \geq \mathbf{P}\left(\forall t \in [T_{Jf+1}, \ldots, T_J], \left\|\widehat{k}_t - \kappa^*\right\| \leq \epsilon\right). \tag{13}$$

The minimum and maximum batch sizes are $b_{\min} = \frac{Bs}{c_{\max}}$ and $b_{\max} = \frac{Bs}{c_{\min}}$, respectively. Therefore,

$$T_{Jf+1} \geq Jfb_{\min} = Jf\frac{Bs}{c_{\max}},$$

$$T_J \leq Jb_{\max} = J\frac{Bs}{c_{\min}}.$$

Using these facts and continuing Eq. 13, we get:

$$
\begin{aligned}
\mathbf{P}\left(\forall j \in [Jf+1, Jf+2, \ldots, J], \left\|\widehat{k}_{T_j} - \kappa^*\right\| \leq \epsilon\right) &\geq \mathbf{P}\left(\forall t \in [T_{Jf+1}, \ldots, T_J], \left\|\widehat{k}_t - \kappa^*\right\| \leq \epsilon\right) \\
&\geq \mathbf{P}\left(\forall t \in [Jfb_{\min}, \ldots, Jb_{\max}], \left\|\widehat{k}_t - \kappa^*\right\| \leq \epsilon\right) \\
&\overset{(a)}{\geq} 1 - \sum_{t=Jfb_{\min}}^{Jb_{\max}} \frac{1}{\epsilon^{4D}} \exp\left\{-\mathcal{O}\left(t\epsilon^8\right)\right\} \\
&\overset{(b)}{\geq} 1 - \sum_{t=Jfb_{\min}}^{Jb_{\max}} \frac{1}{\epsilon^{4D}} \exp\left\{-\mathcal{O}\left(Jfb_{\min}\epsilon^8\right)\right\} \\
&\geq 1 - \sum_{t=Jfb_{\min}}^{Jb_{\max}} \frac{1}{\epsilon^{4D}} \exp\left\{-\mathcal{O}\left(Bf\epsilon^8\right)\right\} \\
&\geq 1 - \frac{Jb_{\max}}{\epsilon^{4D}} \exp\left\{-\mathcal{O}\left(Bf\epsilon^8\right)\right\} \\
&\geq 1 - \frac{B}{\epsilon^{4D}} \exp\left\{-\mathcal{O}\left(Bf\epsilon^8\right)\right\} \\
&\geq 1 - \frac{B}{\epsilon^{4D}} \exp\left\{-\mathcal{O}\left(B^\gamma\epsilon^8\right)\right\} \\
&\to 1,
\end{aligned}
$$

where (a) follows by the union bound and (b) because $t \geq Jfb_{\min}$. $\qquad\square$

## C  Feasible regions

In this section, we derive the feasibility regions for the various policies.

### OMS-ETC

Recall that in OMS-ETC, we first collect $Te$ samples such that $\kappa_{Te} = \text{ctr}\left(\Delta_\psi\right)$. For the remaining $T(1-e)$ samples, the agent can query the data sources with any fraction $\kappa \in \Delta_\psi$. Therefore, the feasible values of $\kappa_T$ are

$$
\begin{aligned}
\tilde{\Delta} &= \left\{\frac{Te\kappa_{Te} + T(1-e)\kappa}{T} : \kappa \in \Delta_\psi\right\} \\
&= \left\{e\kappa_{Te} + (1-e)\kappa : \kappa \in \Delta_\psi\right\}.
\end{aligned}
$$

### OMS-ETG

After $j$ rounds, the selection ratio is denoted by $\kappa_{bj}$. In every round, we collect $b = Ts$ samples. For the batch collected in round $j+1$, the agent can query the data sources with any fraction $\kappa \in \Delta_\psi$.

Therefore, the feasible values of $\kappa_{b(j+1)}$ are

$$\tilde{\Delta}_{j+1}(\kappa_{bj}) = \left\{ \frac{bj\kappa_{bj} + b\kappa}{b(j+1)} : \kappa \in \Delta_\psi \right\}$$

$$= \left\{ \frac{Tsj\kappa_{bj} + Ts\kappa}{Ts(j+1)} : \kappa \in \Delta_\psi \right\}$$

$$= \left\{ \frac{j\kappa_{bj} + \kappa}{(j+1)} : \kappa \in \Delta_\psi \right\}.$$

## OMS-ETC-CS

The agent uses $Be$ budget to uniformly query the available data sources. Let $T_e$ denote the number of samples collected after exploration. We have

$$T_e = \frac{Be}{\kappa_{T_e}^\top c},$$

where $\kappa_{T_e}^\top = \mathrm{ctr}\,(\Delta_\psi)$ and $c$ is the cost vector. With the remaining $B(1-e)$ budget, the agent can collect samples with any fraction $\kappa \in \Delta_\psi$. However, since the data sources can have different costs, the total number of samples $T$ depends on the choice of $\kappa$:

$$T = T_e + \frac{B(1-e)}{\kappa^\top c},$$

for $\kappa \in \Delta_\psi$. Therefore the feasible values of $\kappa_T$ are

$$\tilde{\Delta} = \left\{ \frac{T_e\kappa_{T_e} + (T - T_e)\kappa}{T} : \kappa \in \Delta_\psi \right\}$$

$$= \left\{ \frac{\frac{Be}{\kappa_{T_e}^\top c}\kappa_{T_e} + \frac{B(1-e)}{\kappa^\top c}\kappa}{\frac{Be}{\kappa_{T_e}^\top c} + \frac{B(1-e)}{\kappa^\top c}} : \kappa \in \Delta_\psi \right\}$$

$$= \left\{ \frac{e\left(\kappa^\top c\right)\kappa_{T_e} + (1-e)\left(\kappa_{T_e}^\top c\right)\kappa}{e\left(\kappa^\top c\right) + (1-e)\left(\kappa_{T_e}^\top c\right)} : \kappa \in \Delta_\psi \right\}.$$

## OMS-ETG-FS

Since we collect a fixed number of samples in each round, the feasibility region for OMS-ETG-FS is that same as OMS-ETG:

$$\tilde{\Delta}_{j+1}(\kappa_{bj}) = \left\{ \frac{j\kappa_{bj} + \kappa}{(j+1)} : \kappa \in \Delta_\psi \right\}.$$

## OMS-ETG-FB

Let the selection ratio after $j$ rounds be $\kappa_{T_j}$ where $T_j$ number of samples collected after round $j$: $T_j = \frac{Bsj}{\kappa_{bj}^\top c}$. For the batch collected in round $j+1$, the agent can query the data sources with any fraction $\kappa \in \Delta_\psi$. However, the number of samples collected in round $j+1$ would depend on the choice $\kappa$ due to the cost structure. Therefore the number of samples collected after round $j+1$ is

$$T_{j+1} = T_j + \frac{Bsj}{\kappa^\top c},$$

for $\kappa \in \Delta_\psi$. Hence, the feasible values of $\kappa_{T_{j+1}}$ are

$$\tilde{\Delta}_{j+1}(\kappa_{T_j}) = \left\{ \frac{T_j\kappa_{T_j} + (T_{j+1} - T_j)\kappa}{T_{j+1}} : \kappa \in \Delta_\psi \right\}$$

$$= \left\{ \frac{\frac{Bsj}{\kappa_{bj}^\top c}\kappa_{T_j} + \frac{Bsj}{\kappa^\top c}\kappa}{\frac{Bsj}{\kappa_{bj}^\top c} + \frac{Bsj}{\kappa^\top c}} : \kappa \in \Delta_\psi \right\}$$

$$= \left\{ \frac{j\left(\kappa^\top c\right)\kappa_{T_j} + \left(\kappa_{T_j}^\top c\right)\kappa}{j\left(\kappa^\top c\right) + \left(\kappa_{T_j}^\top c\right)} : \kappa \in \Delta_\psi \right\}.$$

## D  Experiments

### D.1  Linear IV graph

Data from the linear IV graph (Figure 2a) is simulated as follows:

$$
\begin{aligned}
Z &\sim \mathcal{N}\left(0, \sigma_z^2\right), \\
U &\sim \mathcal{N}\left(0, \sigma_u^2\right), \\
X &:= \alpha Z + \gamma U + \epsilon_x, \ \epsilon_x \sim \mathcal{N}\left(0, \sigma_x^2\right), \\
Y &:= \beta X + \phi U + \epsilon_y, \ \epsilon_y \sim \mathcal{N}\left(0, \sigma_y^2\right),
\end{aligned}
$$

where $\epsilon_x$ and $\epsilon_y$ are exogenous noise terms independent of other variables and each other and $U$ is an unobserved confounder. Here $\{\beta, \alpha, \gamma, \phi, \sigma_z^2, \sigma_u^2, \sigma_x^2, \sigma_y^2\}$ are parameters that we set for simulating the data. For the experiment in Section 6.1, we used $\beta = 1, \alpha = 1, \gamma = 1, \phi = 1, \sigma_z = 1, \sigma_u = 1, \sigma_x = 1, \sigma_y = 1$.

The moment conditions used for estimation are

$$
g_t(\theta) = \underbrace{\begin{bmatrix} s_{t,1} \\ s_{t,2} \end{bmatrix}}_{=m(s_t)} \otimes \underbrace{\begin{bmatrix} Z_t(X_t - \alpha Z_t) \\ Z_t(Y_t - \alpha\beta Z_t) \end{bmatrix}}_{=\tilde{g}_t(\theta)}.
$$

The parameter we estimate is $\theta = [\beta, \alpha]^\top$ and $\beta = f_{\text{tar}}(\theta) = \theta_0$.

### D.2  Two IVs graph

Data from the two IVs graph (Figure 2b) is simulated as follows:

$$
\begin{aligned}
Z_1 &\sim \mathcal{N}\left(0, \sigma_{z_1}^2\right), \\
Z_2 &\sim \mathcal{N}\left(0, \sigma_{z_2}^2\right), \\
U &\sim \mathcal{N}\left(0, \sigma_u^2\right), \\
X &:= \alpha_1 Z_1 + \alpha_2 Z_2 + \gamma U + \epsilon_x, \ \epsilon_x \sim \mathcal{N}\left(0, \sigma_x^2\right), \\
Y &:= \beta X + \phi U + \epsilon_y, \ \epsilon_y \sim \mathcal{N}\left(0, \sigma_y^2\right),
\end{aligned}
$$

where $\epsilon_x$ and $\epsilon_y$ are exogenous noise terms independent of other variables and each other and $U$ is an unobserved confounder. For the experiment in Section 6.1, we used $\beta = 1, \alpha = 1, \gamma = 1, \phi = 1, \sigma_z = 1, \sigma_u = 1, \sigma_x = 1, \sigma_y = 1$.

The moment conditions used for estimation are

$$
g_t(\theta) = \underbrace{\begin{bmatrix} s_{t,1} \\ s_{t,2} \end{bmatrix}}_{=m(s_t)} \otimes \underbrace{\begin{bmatrix} (Z_1)_t(Y_t - \beta X_t) \\ (Z_2)_t(Y_t - \beta X_t) \end{bmatrix}}_{=\tilde{g}_t(\theta)}.
$$

The parameter we estimate is $\theta = [\beta]$ and $\beta = f_{\text{tar}}(\theta) = \theta_0$.

### D.3  Confounder-mediator graph

Data from the confounder-mediator graph (Figure 2b) is simulated as follows:

$$
\begin{aligned}
W &\sim \mathcal{N}\left(0, \sigma_w^2\right), \\
X &:= dW + \epsilon_x, \ \epsilon_x \sim \mathcal{N}\left(0, \sigma_x^2\right), \\
M &:= \frac{\beta}{a}X + \epsilon_m, \ \epsilon_m \sim \mathcal{N}\left(0, \sigma_m^2\right), \\
Y &:= aM + bW + \epsilon_y, \ \epsilon_y \sim \mathcal{N}\left(0, \sigma_y^2\right),
\end{aligned}
$$

where $\epsilon_x, \epsilon_m$, and $\epsilon_y$ are exogenous noise terms independent of other variables and each other. For the experiment in Section 6.1, we used $\beta = -0.32, a = 0.33, b = -0.34, d = 0.45, \sigma_w = 1, \sigma_x = 1, \sigma_m = 1, \sigma_y = 1$.

The moment conditions used for estimation are

$$
g_t(\theta) = \underbrace{\begin{bmatrix} s_{t,1} \\ s_{t,1} \\ s_{t,2} \\ s_{t,2} \\ s_{t,2} \\ s_{t,1} \\ s_{t,1} \\ s_{t,1} \\ 1 \end{bmatrix}}_{=m(s_t)} \otimes \underbrace{\begin{bmatrix} X_t(Y_t - bW_t - \beta X_t) \\ W_t(Y_t - bW_t - \beta X_t) \\ X_t(M_t - \frac{\beta}{a}X_t) \\ M_t\left(Y_t - aM_t - \frac{bd\sigma_w^2}{d^2\sigma_w^2 + \sigma_x^2}X_t\right) \\ X_t\left(Y_t - aM_t - \frac{bd\sigma_w^2}{d^2\sigma_w^2 + \sigma_x^2}X_t\right) \\ W_t^2 - \sigma_w^2 \\ W_t(X_t - dW) \\ X_t^2 - (d^2\sigma_w^2 + \sigma_x^2) \end{bmatrix}}_{=\tilde{g}_t(\theta)}.
$$

The parameter we estimate is $\theta = [\beta, a, b, d, \sigma_w^2, \sigma_x^2]^\top$ and $\beta = f_{\text{tar}}(\theta) = \theta_0$.

## D.4  IHDP dataset

To generate semi-synthetic IHDP dataset, we use two covariates: birth weight (denoted by $W_1$) and whether the mother smoked (denoted by $W_2$). The binary treatment is denoted by $X$ and the outcome is denoted by $Y$. The corresponding causal graph is shown in Figure 4a. For every sample of the semi-synthetic dataset, $W_1, W_2$, and $X$ are sampled uniformly at random from the real data. The outcome $Y$ is simulated as follows:

$$
Y := \beta X + \alpha_1 W_1 + \alpha_2 W_2 + \epsilon_y, \ \epsilon_y \sim \mathcal{N}\left(0, \sigma_y^2\right),
$$

where $\epsilon_y$ is an independent exogenous noise term. For the experiment in Section 6.2, we used $\beta = 1, \alpha_1 = 1, \alpha_2 = 0.1, \sigma_y = 1$.

The moment conditions used for estimation are

$$
g_t(\theta) = \underbrace{\begin{bmatrix} 1 - s_{t,2} \\ 1 - s_{t,2} \\ 1 - s_{t,1} \\ 1 - s_{t,1} \\ s_{t,3} \\ 1 - s_{t,2} \\ 1 - s_{t,1} \\ 1 - s_{t,2} \\ 1 - s_{t,1} \\ 1 - s_{t,2} \\ 1 - s_{t,1} \end{bmatrix}}_{=m(s_t)} \otimes \underbrace{\begin{bmatrix} (W_1)_t\left((Y_t - \alpha_1(W_1)_t - \beta X_t) - \alpha_2 d\right) \\ X_t\left((Y_t - \alpha_1(W_1)_t - \beta X_t) - \alpha_2 \tau_2\right) \\ (W_2)_t\left((Y_t - \alpha_2(W_2)_t - \beta X_t) - \alpha_1 d\right) \\ X_t\left((Y_t - \alpha_2(W_2)_t - \beta X_t) - \alpha_1 \tau_1\right) \\ (W_1)_t(W_2)_t - d \\ X(W_1)_t - \tau_1 \\ X(W_2)_t - \tau_2 \\ (W_1)_t^2 - \sigma_{w_1}^2 \\ (W_2)_t^2 - \sigma_{w_2}^2 \\ (Y_t - \alpha_1(W_1)_t - \beta X)^2 - \alpha_2^2\sigma_{w_2}^2 - \sigma_y^2 \\ (Y_t - \alpha_2(W_2)_t - \beta X)^2 - \alpha_1^2\sigma_{w_1}^2 - \sigma_y^2 \end{bmatrix}}_{=\tilde{g}_t(\theta)}.
$$

The parameter we estimate is $\theta = [\beta, \alpha_1, \alpha_2, d, \tau_1, \tau_2, \sigma_w^2, \sigma_y^2]^\top$ and $\beta = f_{\text{tar}}(\theta) = \theta_0$.

## D.5  The Vietnam draft and future earnings dataset

The causal graph for this dataset corresponds to Figure 2a with a binary IV $Z$, binary treatment $X$ and continuous outcome $Y$. In this dataset, $\{Z, X\}$ and $\{Z, Y\}$ are collected from different data sources and thus $\{Z, X, Y\}$ are not observed simultaneously. For our experiment, we only use data from the 1951 cohort.

In the semi-synthetic dataset, we sample $Z$ uniformly at random from the real dataset. The treatment $X$ is generated similarly to a probit model. We first generate an intermediate variable $X^*$ and then use that to generate $X$ as follows:

$$
X^* := \alpha Z + c^* + \epsilon_x, \ \epsilon_x \sim \mathcal{N}(0, 1),
$$
$$
X := \mathbf{1}(X^* > 0),
$$

where $\mathbf{1}$ is the indicator function. To reduce clutter, let $\mu_z = \widehat{\mathbf{P}}(Z = 1) = 0.3425$, $\mu_x^{(1)} = \mathbf{P}(X = 1 | Z = 1)$ and $\mu_x^{(0)} = \mathbf{P}(X = 1 | Z = 0)$. We set the parameters $\alpha$ and $c^*$ such that $\mu_x^{(1)} = 0.2831$ and

$\mu_x^{(0)} = 0.1468$ (these values have been taken from [2, Table 2] to match the empirical distribution):

$$\begin{aligned}
\mu_x^{(0)} &= \mathbf{P}(\mathbf{1}(X^* > 0)|Z = 0) \\
&= \mathbf{P}(c^* + \epsilon_x > 0)) \\
&= \mathbf{P}(\epsilon_x > -c^*)) \\
&= \mathbf{P}(\epsilon_x < c^*)) \\
&= \Phi(c^*), \\
\therefore c^* &= \Phi^{-1}(\mu_x^{(0)}) \\
&= \Phi^{-1}(0.1468) \\
&= -1.050, \\
\mu_x^{(1)} &= \mathbf{P}(\mathbf{1}(X^* > 0)|Z = 1) \\
&= \mathbf{P}(\alpha + c^* + \epsilon_x > 0)) \\
&= \Phi(\alpha + c^*) \\
\therefore \alpha &= \Phi^{-1}(\mu_x^{(1)}) - c^* \\
&= \Phi^{-1}(\mu_x^{(1)}) - \Phi^{-1}(\mu_x^{(0)}) \\
&= \Phi^{-1}(0.2831) - \Phi^{-1}(0.1468) \\
&= 0.4766,
\end{aligned}$$

where $\Phi$ is the cumulative distribution function of the standard normal distribution.

In the real data, we standardize the outcome $Y$ by subtracting its mean and dividing by its standard deviation and thus $\widehat{\mathbf{E}}[Y] = 0$ and $\widehat{\mathrm{Var}}(Y) = 1$. To generate the simulated outcome $Y$, we use $Y := \beta X + \gamma + c_0\epsilon_x + \epsilon_y$, where $\epsilon_y \sim \mathcal{N}(0, \sigma_{\epsilon_y}^2)$. When $c_0 \neq 0$, the noise term $(c_0\epsilon_x + \epsilon_y) \not\perp X$. Thus $c_0$ determines the extent of the confounding between $X$ and $Y$.

We now describe how we set $\beta$ and $\gamma$. Since $E[Y] = 0$, we have

$$\begin{aligned}
\gamma &= -\beta\mathbf{E}[X] \\
&= -\beta\left(\mu_x^{(0)}(1 - \mu_z) + \mu_x^{(1)}\mu_z\right) \\
&= -0.1934\beta.
\end{aligned}$$

Using the covariance of $Y$ and $Z$, we have

$$\begin{aligned}
\mathrm{Cov}(Y, Z) &= \mathbf{E}[YZ] \\
&= \beta\mathbf{E}[ZX] + \gamma\mathbf{E}[Z] \\
&= \beta\left(\mathbf{E}[ZX] - \mathbf{E}[X]\mathbf{E}[Z]\right) \\
&= \beta\left(\mathbf{E}[Z\mathbf{1}(\alpha Z + c^* + \epsilon_x > 0)] - \mathbf{E}[X]\mathbf{E}[Z]\right) \\
&= \beta\left(\mathbf{E}[Z\mathbf{E}[\mathbf{1}(\alpha Z + c^* + \epsilon_x > 0)|Z]] - \mathbf{E}[X]\mathbf{E}[Z]\right) \\
&= \beta\left(\mathbf{E}[Z\mathbf{E}[\mathbf{1}(\epsilon_x > -(\alpha Z + c^*))|Z]] - \mathbf{E}[X]\mathbf{E}[Z]\right) \\
&= \beta\left(\mathbf{E}[Z\Phi(\alpha Z + c^*)] - \mathbf{E}[X]\mathbf{E}[Z]\right) \\
&= \beta\left(\Phi(\alpha Z + c^*)\mu_z - \mathbf{E}[X]\mathbf{E}[Z]\right) \\
&= \beta\mu_z\left(\mu_x^{(1)} - \mathbf{E}[X]\right).
\end{aligned}$$

Therefore, we set $\beta$ and $\gamma$ as

$$\beta = \frac{\widehat{E}[YZ]}{\mu_z\left(\mu_x^{(1)} - \mathbf{E}[X]\right)} = -0.4313,$$

$$\gamma = -0.1934\beta = 0.0834.$$

Now we describe how we set $c_0$ and $\sigma_{\epsilon_y}^2$. For this, we use the variance of $Y$:

$$\mathrm{Var}(Y) = 1 = \beta^2\mathrm{Var}(X) + c_0^2\sigma_{\epsilon_y}^2 + 2\beta c_0\mathbf{E}[X\epsilon_x]. \tag{14}$$

We have

$$\mathrm{Var}(X) = \mathrm{Var}[\mathbf{E}(X|Z)] + \mathbf{E}[\mathrm{Var}(X|Z)]$$

$$= \mathrm{Var}\left(Z\mu_x^{(1)} + (1-Z)\mu_x^{(0)}\right) + \mu_z\mu_x^{(1)}(1-\mu_x^{(1)}) + (1-\mu_z)\mu_x^{(0)}(1-\mu_x^{(0)})$$

$$= \mu_z(1-\mu_z)(\mu_x^{(1)} - \mu_x^{(0)})^2 + \mu_z\mu_x^{(1)}(1-\mu_x^{(1)}) + (1-\mu_z)\mu_x^{(0)}(1-\mu_x^{(0)})$$

$$= 0.1560,$$

$$\mathbf{E}[X\epsilon_x] = \mathbf{E}[\mathbf{E}[\mathbf{1}(Z\alpha + c^* + \epsilon_x > 0)\epsilon_x|Z]]$$

$$= \mathbf{E}[\mathbf{E}[\mathbf{1}(\epsilon_x > -(Z\alpha + c^*))\epsilon_x|Z]]$$

$$\overset{(a)}{=} \mathbf{E}_Z\left[\int_{-(Z\alpha+c^*)}^{\infty} xf(x)dx\right]$$

$$= \mathbf{E}\left[\frac{1}{\sqrt{2\pi}}\exp\left\{\frac{-(Z\alpha + c^*)^2}{2}\right\}\right]$$

$$= \frac{1}{\sqrt{2\pi}}\left[\exp\left\{\frac{-(c^*)^2}{2}\right\}(1-\mu_z) + \exp\left\{\frac{-(\alpha + c^*)^2}{2}\right\}\mu_z\right]$$

$$= 0.2670,$$

where in (a), $f(x)$ is the probability density function of the standard normal distribution. We set $c_0 = 0.5$ and using Eq. 14, we get $\sigma_{\epsilon_y}^2 = 0.6058$.

To summarize, the data is generated as follows:

$$Z \sim \mathrm{Bernoulli}(\mu_z),$$

$$X^* := \alpha Z + c^* + \epsilon_x, \ \epsilon_x \sim \mathcal{N}(0,1),$$

$$X := \mathbf{1}(X^* > 0),$$

$$Y := \beta X + \gamma + c_0\epsilon_x + \epsilon_y, \ \epsilon_y \sim \mathcal{N}(0, \sigma_{\epsilon_y}^2),$$

where $\mu_z = 0.3424, \alpha = 0.4766, c^* = -1.0502, \beta = -0.4313, \gamma = 0.0834$, and $\sigma_{\epsilon_y}^2 = 0.6058$.

The moment conditions used for estimation are

$$g_t(\theta) = \underbrace{\begin{bmatrix} s_{t,1} \\ s_{t,1} \\ s_{t,2} \\ s_{t,2} \end{bmatrix}}_{=m(s_t)} \otimes \underbrace{\begin{bmatrix} Z_t(Y_t - \mu_1) \\ (1 - Z_t)(Y_t - \mu_0) \\ Z_t(X_t - \tau_1) \\ (1 - Z_t)(X_t - \tau_0) \end{bmatrix}}_{=\tilde{g}_t(\theta)}.$$

The parameter we estimate is $\theta = [\mu_1, \mu_0, \tau_1, \tau_0]$ and the target parameter is $\beta = f_{\mathrm{tar}}(\theta) = \frac{\mu_1 - \mu_0}{\tau_1 - \tau_0}$.