# OpenReview forum: "Efficient Online Estimation of Causal Effects by Deciding What to Observe"
_NeurIPS.cc/2021/Conference — NeurIPS 2021 Spotlight_

### Official Review · Reviewer_pVCh · 2021-07-15

**Rating:** 8
**Confidence:** 2

**Summary:**

The asymptotic property of GMM estimator is proposed by some previous papers. In this paper, from the theoretical aspect, the authors generalize that to the setting with multiple domains and apply the results in the online estimation of causal effects. They evaluate the regret under three online policies to select which dataset to observe.

**Limitations And Societal Impact:**

Yes

**Main Review:**

After rebuttal: the authors have addressed my quesions. The paper is very interesting and I am happy to see it accepted. I increase my score to 8.

------------------------------------
At first, I want to say that I am not familiar with asymptotic theory. Although I read the proof and some related papers [15,27], I do not feel quite qualified to judge at the theoretical level. Hence it is hard for me to give some suggestions from theoretical viewpoint. Hence I mainly focus on the algorithm.

The main reason that I give a borderline score is that the problem tackled seems not reasonable. This makes me think the method right now is solid, but does not contribute a lot to causality literature, especially in a causal effect estimation task.

It seems that in reality, the optimal policy is always as [0,0,\cdots,0,1,0,\cdots,0] (It implies one dataset is better then the others). The estimator under different datasets have different asymptotic distributions. In this case, the optimal policy is to select the dataset with which the estimator has the least asymptotic variance. Hence it seems the main problem is to estimate the asymptotic variance for each dataset. Although the authors propose a more general method, which could be applied no matter what the optimal policy is like, it seems the contribution does not have a strong connection with causal effect estimation task. Could the authors further provide an example in reality when the optimal policy is not as [0,0,\cdots,0,1,0,\cdots,0]?


In addition, there are a few suggestions, questions, or reminders of typos:
-The authors had better propose the explanation of symbol \tilde{g_t}(\theta) in Line 115.
-Line 181: Appendix D -> Appendix C
-I do not understand the equation on Line 170. Why is the order of difference constant when the policy is fixed?
-In reality, it seldom holds that the parameter could be identified by each dataset. As said by the example of doctor on Line 19,  the different datasets are possible from different interventions (tests). In this case, it is hard to guarantee that the parameter is identifiable in each dataset. Hence I think it will be better if the authors could mention that they discard the dataset by which the parameter is unidentifiable.



I look forward to the authors' rebuttal. And I will be happy to increase my score if the authors could provide some persuasive opinions, no matter by correcting me or responding to my questions above.

**Time Spent Reviewing:**

4

---

> ### Author Response · Authors · 2021-08-10
> **Responded to primary concern regarding optimal policy always on the corner of the simplex**
>
> We thank the reviewer for their feedback. Below we respond to each of their concerns. We believe that your primary concern centers on when we might find a problem for which the optimal policy does not sit on the corner of the simplex ($[0,0,\cdots,0,1,0,\cdots,0]$). Below, we show that these cases are surprisingly common, both in the prior literature and in the very examples addressed in this paper (see more detail below). We hope that our explanations alleviate your concerns and promise to improve the exposition around these points in the camera-ready version.
>
>
> *Re problem tackled seems not reasonable:*
>
> Typically, in causal inference, the dataset is assumed to have been provided in advance and this dataset is then used to estimate the causal effect. However, in many important applications (e.g., medical settings, surveys), the experimenter has control over the data collection. However, this part is usually ignored while performing causal inference and the existing techniques focus on identification and estimation. Our work takes initial strides towards endogenizing the data collection decisions with the estimation of causal effects. We aim to couple data collection with estimation and thus our methods should be relevant to a broad class of causal inference settings where the practitioner can choose which variables to collect.
>
> Similar settings have been studied in previous works as the data fusion problem [1]. [1,2, 3] study the identifiability of causal effects in cases where different subsets of variables are observed in different datasets. [4, 5] study finding the optimal covariate adjustment set when multiple sets are available. Unlike our setting, these works assume that the datasets are fixed and available in advance. For many causal graphs (including the ones in our experiments), the optimal choice of variables depends on the underlying parameters of the graph itself and this motivates us to estimate the causal effects in an online fashion.
>
> *Re “the optimal policy is always as [0,0,\cdots,0,1,0,\cdots,0]”:*
>
> There certainly are some cases where the optimal policy is on the corner and in these cases the best that one can do is to sample as much as possible from the data source with the lowest variance. For an example, see the two instrumental variables case (Figure 2b). However, there are also broad classes of problems where the optimal policy selects a mix of different data sources (in other words, the oracle ratio $\kappa^*$ lies in the interior of the simplex).  There are three main scenarios where the optimal policy would be in the interior:
>
> 1. When multiple datasets are required to identify the causal effect. Consider the instrumental variable (IV) graph described in Example 1 where the data sources are $\Psi = \{ \{ Z, X \}, \{ Z, Y \} \}$ and at each time, we can collect either $\{ Z, X \}$ or $\{ Z, Y \}$. In this case, we need a mix of both these data sources in order to identify the causal effect. If we only collect one of the two, the effect can no longer be identified. Therefore, in this case, the optimal policy will lie in the interior of the simplex. You rightly point out later in your review that “it is hard to guarantee that the parameter is identifiable in each dataset”. Our methods are applicable even when a single dataset does not identify the ATE. The IV example is an instance where one dataset cannot identify the ATE and thus we must collect a mix of them (thus the optimal policy is not $[0, \cdots,1, \cdots, 0]$). In this case, we will \emph{not} discard either dataset but use both of them.
>
>
> 2. Even in over-identified causal graphs where each dataset can identify the causal effect, it is possible that a combination of multiple datasets is optimal rather than only collecting one of them. In other words, it is not a matter of picking the dataset with the lowest asymptotic variance but rather of deciding the optimal ratio of the two datasets. Recently, [6] discussed the confounder-mediator graph (Figure 2c) where both the backdoor (using confounders) and frontdoor (using mediators) adjustments are valid. They show that it is possible that collecting a mix of confounders and mediators leads to lower variance than just using one strategy.
>
> 3. The cost structure might make one of the datasets too expensive. Thus even if a dataset has lower asymptotic variance, it might be costlier than the other dataset. Under a fixed budget, it can be optimal to collect some fraction of the other dataset (since you can get more samples for the same cost) and thus the optimal policy can be in the interior and not to only collect the better dataset. We provide two examples of this in our experiments. In this confounder-mediator graph (Figure 2c) and the IHDP dataset (Figure 4a), different data sources have different costs and the optimal policies are  $\kappa^* \approx [0.15, 0.85]^\top$ and  $\kappa^* \approx [0.59, 0, 0.41]^\top$, respectively.
>
> Thus there can be many situations where the optimal policy can involve collecting a mix of different data sources. Our methods offer a unified approach to dealing with all of these cases.
>
>
> *Re better explanation of $g_t(\theta)$:*
>
> In our setup, the moment functions can be written as $g_t(\theta) := m(s_t) \otimes \tilde{g}_t(\theta)$ where $m(s_t)$ represents the choice of data source that is selected at time $t$ (it can be thought of as a binary mask vector indicating which moments are selected) and $\tilde{g}_t(\theta)$ represents the i.i.d. empirical moments computed using the collected data samples. We promise to improve the explanation of $g_t(\theta)$ in the camera-ready version.
>
>
> *Re Line 170: order of difference constant:*
>
> Apologies, we agree that the big-$O$ notation here makes it harder to interpret. Big-$\Theta$ is more appropriate here. We were trying to say that the regret of any fixed policy (except the oracle) will be some non-zero constant. This is because the oracle policy has the lowest possible asymptotic MSE and hence every other fixed policy will have non-zero regret.
>
>
>
>
> [1] Bareinboim, E., & Pearl, J. (2016). Causal inference and the data-fusion problem. Proceedings of the National Academy of Sciences, 113(27), 7345-7352.
>
> [2] S. Lee, and E. Bareinboim. Causal effect identifiability under partial-observability. In International Conference on Machine Learning. PMLR, 2020
>
> [3] S. Tikka, A. Hyttinen, and J. Karvanen. Causal effect identification from multiple incomplete data sources: A general search-based approach. arXiv preprint arXiv:1902.01073, 2019.
>
> [4] Leonard Henckel, Emilija Perkovi´c, and Marloes H Maathuis. Graphical criteria for efficient total effect estimation via adjustment in causal linear models. arXiv preprint
> arXiv:1907.02435, 2019.
>
> [5] A. Rotnitzky, and E. Smucler. (2020). Efficient Adjustment Sets for Population Average Causal Treatment Effect Estimation in Graphical Models. JMLR, 21, 188-1.
>
> [6] Gupta, S., Lipton, Z. C., & Childers, D. (2021). Estimating treatment effects with observed confounders and mediators. In Uncertainty in Artificial Intelligence.

---

> > ### Comment · Reviewer_pVCh · 2021-08-12
> > **Discussion**
> >
> > Thank you very much for your response. It addresses my questions. The paper is very interesting. I like it! I am happy to see the paper accepted and I increase my score to 8.
> >
> > Best Regards

---

### Official Review · Reviewer_XDyk · 2021-07-16

**Rating:** 7
**Confidence:** 4

**Summary:**

This paper proposes online moment selection, an approach for choosing which data sources to draw from in an online setting. The goal is estimation of causal effects, the structural causal model is encoded as moment conditions, and the data sources drawn from are chosen to minimize the variance of the estimate. Explore then commit and explore then greedy approaches are suggested and analyzed. A limitation is that the structural equations need to be parametric and have a clear noise term. Improvement is shown over simple fixed policy baselines.

**Limitations And Societal Impact:**

no social impact

**Main Review:**

I found the problem interesting and the results good. I would have liked to see more discussion of the many assumptions and some exploration of whether they hold in practice and a theoretical discussion of under what types of causal models we can expect them to hold, e.g. the uniqueness assumptions etc.

Line 163: I assume it is the case that this oracle fixed policy is also optimal across adaptive policies, if so this is important to clarify as the regret is being defined for an active policy.

Minor: may be of interest to include related work on active learning of causal models, e.g. (Squires et al).

**Time Spent Reviewing:**

NA

---

> ### Author Response · Authors · 2021-08-10
> **Explained the uniqueness assumption, promise to add more discussion of assumptions and conditions for our theorems**
>
> We thank the reviewer for their constructive suggestions, and briefly touch on the main points:
>
> *Re more discussion of the assumptions … whether they hold in practice:*
>
> We agree that in their current state, the assumptions and conditions for the theorems are difficult to understand. We will add more text to informally explain these assumptions and provide more intuition. We would like to point out that our theoretical results cover a broad class of models: bounded and linear moments (with sub-Exponential data) satisfy them and thus the theoretical results apply to the graphs used in our experiments.
>
> *Re uniqueness assumption:*
>
> The uniqueness assumption in Assumption 3 is required to make the choices of the datasets meaningful. It will be violated if two different data source choices lead to the exact same variance of the GMM estimator. As an example, consider the graph with two instrumental variables (Figure 2b). This assumption will be violated if both the instruments were equally good and thus any combination of the two will lead to the same variance (i.e., the selection is no longer meaningful). This is a weak assumption and we expect it to be satisfied in practice.
>
> *Re oracle also optimal across adaptive policies:*
>
> Yes, that is correct. We will clearly mention this in the text.
>
> *Re related work on active learning of causal models:*
>
> Thank you for the references. We will add them to the related works.

---

> > ### Comment · Reviewer_XDyk · 2021-08-12
> > **repy**
> >
> > Thanks for the clarifications and incorporating the recommendations. I continue to be positive about the paper.

---

### Official Review · Reviewer_vjWD · 2021-07-16

**Rating:** 6
**Confidence:** 4

**Summary:**

The paper proposes online moment selection (OMS), a framework in which
structural assumptions are encoded as moment conditions. This
algorithm allows users to actively acquire data (variables) given the
very moments that identify the functional of interest. The paper shows
that there is substantial empirical gains in doing so.

**Limitations And Societal Impact:**

The paper discuss the limitations in terms of the assumptions it require.

**Main Review:**

The paper proposes online moment selection (OMS), a framework in which
structural assumptions are encoded as moment conditions. This
algorithm allows users to actively acquire data (variables) given the
very moments that identify the functional of interest. The paper shows
that there is substantial empirical gains in doing so.

The theoretical and empirical results of the paper make sense. I have
a few major comments.

One comment is on the potential bias due to actively collecting data.
In particular, several works have identified that actively acquired
data / variables have bias due to the active nature of the procedure.
[*] Does the proposed algorithm suffer from similar bias too? Why or
why not? Is there any correction procedure that would avoid this
issue?

[*] Nie, X., Tian, X., Taylor, J., & Zou, J. (2018, March). Why
adaptively collected data have negative bias and how to correct for
it. In International Conference on Artificial Intelligence and
Statistics (pp. 1261-1269). PMLR.

Farquhar, S., Gal, Y., & Rainforth, T. (2020, September). On Statistical Bias In Active Learning: How and When to Fix It. In International Conference on Learning Representations.

The other comment is that the theoretical results of the paper focus
on zero asymptotic regret, which is a fairly weak guarantee. Loosely,
it says eventually we will converge to the truth. Do we know how fast
the algorithm converge to the truth? Can we bound the finite step
regret? Does the algorithm admit optimal finite horizon regret? How
does regret grow with the number of time steps?


**Time Spent Reviewing:**

10

---

> ### Author Response · Authors · 2021-08-10
> **Addressed concerns regarding bias due to adaptive collection, convergence rates, and finite-sample results for the regret**
>
> Thank you for this constructive feedback. We briefly respond to your concerns regarding our theoretical contributions.
>
> *Re potential bias due to actively collecting data:*
>
> Thanks for raising this interesting point and for providing the two references. Our analysis shows that our estimators weakly converge to a Gaussian and thus are *asymptotically unbiased*, i.e, the bias vanishes as $T \to \infty$.
>
> The GMM estimator is known to have finite-sample bias in many cases (including the instrumental variable case) but this bias is higher-order and thus is asymptotically negligible (a reference is [3] where they discuss higher-order asymptotic properties of GMM). Like the bandit setting you pointed to in [1], there may be some bias due to adaptive collection which manifests in higher-order terms. We examined our experimental data and found that this bias is indeed negligible. Even for small sample sizes, the MSE is dominated by the variance and the bias contributes less than $0.1$% to it. Thus this bias is not a concern in practice.
>
> On the other hand, our methods may not suffer from the active learning bias discussed in [2] where the features $x$ are actively selected. By contrast, in our setting, the choice of which variables to collect is dependent on the history but the samples from the data sources are i.i.d.
>
>
>
> *Re how fast the algorithm converges to the truth:*
>
> Our analysis, which relies on a martingale Central Limit Theorem (CLT), shows that the adaptive estimators, scaled by $\sqrt{T}$, converge weakly to a Gaussian distribution.
> For the asymptotic regret, we scale the estimator by $\sqrt{T}$ and thus finding a convergence rate for the regret would require finding the rate at which the distribution converges to the Gaussian. CLTs only allow us to prove weak convergence but do not provide convergence rates. This would potentially require proving a martingale Berry-Esseen bound for these estimators, which requires stronger assumptions (such as additional moments) which might not hold for GMM estimators (e.g., the instrumental variable case).
>
>
> *Re finite-sample results for the regret:*
>
> Our current proof technique does not give us finite-sample regret bounds. To bound finite-sample regret, we would need to derive bounds on the finite-sample MSE of the GMM estimators under adaptive data collection. The MSE for GMM estimators is known to not exist in many cases (e.g. the instrumental variable case where not even the first moment exists [4]). Therefore, analyzing finite-sample regret would likely require restricting to cases that rule out such examples or require an alternative formulation of regret not based on MSE.
>
> We were able to derive finite-sample tail bounds for the GMM estimator with non-i.i.d. data (see Lemma 1) using uniform martingale laws (which is also a novel result). However, the martingale proof renders the bound policy agnostic and so it cannot be used to compare the regret of two different policies.
>
>
>
>
> [1] Nie, X., Tian, X., Taylor, J., & Zou, J. (2018, March). Why adaptively collected data have negative bias and how to correct for it. In International Conference on Artificial Intelligence and Statistics (pp. 1261-1269). PMLR.
>
> [2] Farquhar, S., Gal, Y., & Rainforth, T. (2020, September). On Statistical Bias In Active Learning: How and When to Fix It. In International Conference on Learning Representations.
>
> [3] Newey, W. K., & Smith, R. J. (2004). Higher order properties of GMM and generalized empirical likelihood estimators. Econometrica, 72(1), 219-255.
>
> [4] Nelson, C. R., & Startz, R. (1990). Some further results on the exact small sample properties of the instrumental variable estimator. Econometrica: Journal of the Econometric Society, 967-976.

---

> > ### Comment · Reviewer_vjWD · 2021-08-31
> > **Thank you for your response!**
> >
> > Thank you for your response to the reviews. My (positive) evaluation of the paper stays the same.

---

### Official Review · Reviewer_vPND · 2021-07-17

**Rating:** 7
**Confidence:** 3

**Summary:**

Given multiple data sources, each related to a subset of variables, authors study the problem of deciding which data source to sample from at each time step. Then, by using the samples, the goal is to estimate the model parameters as good as possible. The proposed framework is based on the generalized method of moments. Their contributions are as follows:

1. In Section 4, it is assumed that each instance has the same cost across all data sources. Then, authors first show that any fixed policy that differs from the optimal one will attain a constant asymptotic regret. To overcome this issue, two adaptive methods are proposed OMS-ETC and OMS-ETG. Here, OMS-ETC resembles the classical explore-then-commit algorithm in the context of bandits, where there is an exploration step to estimate the model parameters and then the model commits to a data source that minimizes the asymptotic variance of the target parameter. In contrast, OMS-ETG keeps updating the model parameters as more data is gathered. Under certain conditions, authors show that both adaptive methods will attain a zero asymptotic regret.

2. In Section 5, both adaptive methods are modified as to attain zero regret in the case where instances have different costs across data sources.

3. Experiments are provided in Section 6, where authors depict the behavior of the regret for different causal DAGs. Semi-synthetic dataset are also provided to showcase the usability of their method.

**Limitations And Societal Impact:**

Yes

**Main Review:**

Here I present my review along the axes from the NeurIPS reviewing guidelines:

**Originality**

- Are the tasks or methods new?
  - Yes. I believe the proposed setting is new as it differs from the data fusion problem.

- Is the work a novel combination of well-known techniques?
  - Yes. The methods are motivated by similar ideas applied in online settings.

- Is it clear how this work differs from previous contributions?
  - Yes. The main contrast starts with the setting presented in the paper, as it can be regarded as a "online data fusion" problem.

- Is related work adequately cited?
  - Yes, authors compare their methods against relevant work.


**Quality**

- Is the submission technically sound?
  - Yes, the theoretical analysis relies heavily on martingale theory and leverages existing concentration and asymptotic results from other settings.

- Are claims well supported (e.g., by theoretical analysis or experimental results)?
  - Yes. Proofs are given in the appendix, and experiments effectively support the theoretical claims.

- Are the methods used appropriate?
  - Yes, given the non-iid nature of moment samples, martingale is an appropriate tool for the analysis.

- Is this a complete piece of work or work in progress?
  - This work is complete.

- Are the authors careful and honest about evaluating both the strengths and weaknesses of their work?
  - Yes, authors state the conditions/assumptions required for their methods.


**Clarity**

- Is the submission clearly written?
  - Yes. I particularly appreciate Example 1, which helped grasp the setting faster.

- Is it well organized?
  - Yes, I was able to follow the main results without issues.

- Does it adequately inform the reader?
  - Yes, several details are provided to reproduce the results.


**Significance**

- Are the results important?
  - Yes, the setting along with the algorithms provided are of interest to the machine learning community. In the introduction, authors provide a compelling story to motivate the setting.

- Are others (researchers or practitioners) likely to use the ideas or build on them?
  - Very likely in my opinion. The setting itself might get the attention of other researchers to propose alternative methods or improve upon the proposed ones.

- Does the submission address a difficult task in a better way than previous work?
  - As the setting differs to existing ones, there is no direct comparison to other methods.

- Does it advance the state of the art in a demonstrable way?
  - The methods presented can be regarded as the first set of results for the setting under study.

# Questions / Comments

I believe the paper discusses an interesting setting, where two non-trivial algorithms (motivated by existing methods from the online learning community) are proposed. My questions relate to the assumptions required to carry out the regret analyses of the methods.

1. The identifiability assumption given in Assumption 2 was initially hard to follow to me. Can authors provide more intuition/interpretation on this assumption?

2. Do authors have any comments about the necessary conditions for their methods?

3. I understand that the reason for the lack of comparison against other methods is due to the setting. Still, I wonder if there is a possibility to (even naively) apply existing methods for ATE estimation under this setting.

---
I thank the authors for their response. I remain positive about the paper and still recommend acceptance.

**Time Spent Reviewing:**

6

---

> ### Author Response · Authors · 2021-08-10
> **More details on identifiability, necessary conditions for our methods, and comparison against existing methods**
>
> Thank you for your considerate review. We address your three questions below:
>
>
> *Re Q1  (identifiability assumption in Assumption 2 hard to follow):*
>
> By identifiability here, we mean that there is a unique value of the relevant model parameters that is consistent with the observed data and (excepting some fine print) this generally means that we can develop consistent estimators that converge to the true value of the parameters as we collect more and more data. In our setup, there are two basic conditions to ensure identifiability: (i) the available data sources must be sufficient to uniquely pin down the parameters given unlimited data from each; (ii) for each moment, our policy must collect a non-negligible fraction of samples from data sources from which it can be estimated. In most of our examples, requirement (ii) simply means that our policy must collect a non-negligible fraction of samples from each data source.
>
> We will improve the exposition around Assumption 2 in the camera-ready version.
>
>
>
> *Re Q2 (“Do authors have any comments about the necessary conditions for their methods?”):*
>
> While we aimed to present sufficient conditions that were transparent, easy to check, and applicable to a broad class of problems, they may not all be necessary to run our methods and obtain consistent estimators. Some of these conditions might be stronger than necessary (e.g., Condition (iv) of Proposition 2 can be weakened to require only two moments instead of $(2 + \delta)$), however, weakening others may impact the convergence rates. For example, it might be possible to use weaker tails than sub-Exponential tails at the cost of slower rates in the finite-sample inequality (Lemma 1).
>
>
>
> *Re Q3 (Lack of comparison against existing methods):*
>
> We compare our adaptive policies to certain naive fixed policies. For example, in the confounder-mediator experiment (Figure 3c) and the Vietnam earnings experiment (Figure 4c), we show that the policy `fixed_all` that queries each data source equally performs poorly. Similarly, for the IHDP experiment (Figure 4b), we show that a policy that naively collects all variables also does poorly. These fixed policies serve as reasonable baselines because this is what practitioners typically do in practice. However, since the setting is new, we did not find other adaptive policies to compare against our methods.

---

### Author Response · Authors · 2021-08-10
**General reply**

We would like to thank all four reviewers for thoughtful feedback and criticism. We are glad to see that the majority of reviewers recommend acceptance and found the paper to be clearly written (R-vPND), the setting novel (R-vPND), empirical gains substantial (R-vjWD), “results good” (R-XDyk), and the method “solid” (R-pVCh). We are also grateful to the reviewers for actionable feedback and address each reviewer’s concerns in the respective threads.

---

### Decision · Program_Chairs · 2021-09-27

**Decision:**

Accept (Spotlight)

**Comment:**

The authors propose a methodology for estimating statistical and causal parameters in an online setting in which multiple available sources of data contain information only on a subset of relevant variables. Repeatedly, an agent much choose which data source to interact with in order to as efficiently as possible estimate the desired parameter. Conditions are given under which the proposed estimators (OMS-ETC, OMS-ETG) achieve vanishing regret, both in uniform and non-uniform cost settings. The approach is evaluated on  synthetic and real-world data.

The reviewers all recognised the novelty of the setting and the value of the proposed algorithm and analysis. There were no major questions or issues remaining after the discussion period, and all four recommended accepting the paper. I believe the unconventional take on the causal estimation problem could be of interest to the wider NeurIPS community.